# Genomic and Transcriptomic Research in the Discovery and Application of Colorectal Cancer Circulating Markers

**DOI:** 10.3390/ijms241512407

**Published:** 2023-08-03

**Authors:** Anastasia A. Ponomaryova, Elena Yu. Rykova, Anastasia I. Solovyova, Anna S. Tarasova, Dmitry N. Kostromitsky, Alexey Yu. Dobrodeev, Sergey A. Afanasiev, Nadezhda V. Cherdyntseva

**Affiliations:** 1Cancer Research Institute, Tomsk National Research Medical Center of the Russian Academy of Sciences, 634009 Tomsk, Russia; 2Institute of Cytology and Genetics of the Siberian Branch of the Russian Academy of Sciences, 630090 Novosibirsk, Russia; 3Department of Engineering Problems of Ecology, Novosibirsk State Technical University, 630087 Novosibirsk, Russia; 4Department of Biochemistry, Medico-Biological Faculty, Siberian State Medical University, 634050 Tomsk, Russia; 5Faculty of Chemistry, National Research Tomsk State University, 634050 Tomsk, Russia

**Keywords:** colorectal cancer, diagnosis, prognosis, prediction of therapy response, circulating DNAs, circulating RNAs, detection of circulating markers

## Abstract

Colorectal cancer (CRC) is the most frequently occurring malignancy in the world. However, the mortality from CRC can be reduced through early diagnostics, selection of the most effective treatment, observation of the therapy success, and the earliest possible diagnosis of recurrences. A comprehensive analysis of genetic and epigenetic factors contributing to the CRC development is needed to refine diagnostic, therapeutic, and preventive strategies and to ensure appropriate decision making in managing specific CRC cases. The liquid biopsy approach utilizing circulating markers has demonstrated its good performance as a tool to detect the changes in the molecular pathways associated with various cancers. In this review, we attempted to brief the main tendencies in the development of circulating DNA and RNA-based markers in CRC such as cancer-associated DNA mutations, DNA methylation changes, and non-coding RNA expression shifts. Attention is devoted to the existing circulating nucleic acid-based CRC markers, the possibility of their application in clinical practice today, and their future improvement. Approaches to the discovery and verification of new markers are described, and the existing problems and potential solutions for them are highlighted.

## 1. Introduction

Colorectal cancer (CRC) is a dangerous disorder, which is characterized by a very high frequency and mortality [1]. Currently, the only workable approach to reduce the CRC-related mortality is the earliest possible diagnosis providing a higher probability for the illness to be treated. The 5-year survival rates directly correlate with the stage when the disease is diagnosed; in particular, the survival rate is considerably higher for the subjects diagnosed at I–IIa CRC stages as compared with later course [2]. The adequate care and screening are most important for the survival rate of CRC subjects, and an early surgical intervention currently gives the best results in the survival of CRC patients.

The ever-increasing evidence suggests that CRC is a heterogeneous and complex disease [3]. The corresponding sequence of events is well known: the process starts from the abnormal crypt proliferation (hyperplasia) via the development of adenomas to carcinomas and eventual metastatic carcinoma; thus, CRC advances in a stepwise manner and the disease progression often takes many years [4]. The predisposition to CRC is to a great degree determined by molecular genetic factors as well as the initiation of the disease, its progression, and metastasizing [5]. The community of experts has recently proposed to create a consensus system for the subtyping of CRC utilizing the cancer-associated gene expression signatures. A bioinformatics analysis and molecular annotation relying on a large set of CRC cohort studies allowed a molecular classification system to be constructed; this system is able to classify most tumors into four robust subtypes that correlate with the disease outcome: namely, consensus molecular subtypes (CMSs), comprising CMS1 (MSI immune), CMS2 (canonical), CMS3 (metabolic), and CMS4 (mesenchymal) [6]. Thanks to biopsies, the attention has been focused on the somatic mutations that occur in several genes at different stages of disease progression. In addition, smoking, diet, lifestyle, and other nongenetic factors are regarded as important contributors to the risk of CRC development. Such factors provide or enhance the conditions for the malignant transformation of somatic cells damaging the genes responsible for the maintenance of genetic stability. Somatic mutations; epigenetic events, including altered DNA methylation; changes in long non-coding RNA expression, changing key gene expression patterns; expression regulation by aberrant non-coding RNA levels; and post-translational modifications at the stage of protein maturation are among the most common molecular events triggered by environmental factors. These changes affect either stability or expression levels of key oncoproteins and tumor suppressor proteins. A comprehensive look at the contribution of molecular genetic factors to the CRC development is necessary to elucidate the pathogenesis route and to improve the diagnostic, therapeutic, and preventive strategies.

The state-of-the-art protocols for the early detection of CRC are insufficiently effective. The most frequently used of them are endoscopy and the fecal occult blood test (FOBT). The latter test is simple, inexpensive, and minimally invasive but, unfortunately, it is poorly sensitive to the early CRC stages [7]. The circulating blood proteome was assayed for the CRC protein markers, such as carbohydrate antigen 19-9 (CA19-9) and carcinoembryonic antigen (CEA); however, both emerged to be insufficiently sensitive and specific, especially for early CRC, and they seem to be insufficient to warrant their widespread use [7]. Therefore, it is most relevant to design new methods and search for new biomarkers to enable the widespread screening of early CRC events.

Circulating cell-free nucleic acids (cfNAs) are a new class of putative biomarkers found in body fluids potentially applicable to detect the disease and improve the CRC outcomes. The cell-free nucleic acids in plasma or serum are advantageous as a minimally invasive diagnostic and prognostic tool for CRC subjects [8,9] (Figure 1). The nucleic acids obtained from biological fluids offer a more “representative” source of biomarkers than biopsy material because of the presence of molecules from different tumor clones and the tumor microenvironment. Somatic mutations and indels, aberrant DNA methylation, aberrant coding and non-coding RNA expression, and other changes have been detected in the blood plasma and serum of cancer patients and have been intensively studied in the last few decades. The accumulation of a considerable amount of omics data has led to the rapid development of new approaches to the mathematical processing of “big data” using the algorithms of artificial intelligence, such as machine learning and deep learning, to advance the methods of screening, early diagnosis, precise prediction, and selection of therapy for cancer diseases, including CRC [10].

However, despite considerable efforts, only a handful of tests based on cfNA have made it into clinical practice. One of them is EpiproColon (Epigenomics AG, Berlin, Germany), detecting the *SEPTIN9* methylation in blood plasma cfDNA; it was approved for commercial application by the United States Food and Drug Administration (FDA) as a diagnostic system for CRC. A low success rate of the systems based on cfNA is explainable with a number of problems associated with the discovered circulating biomarkers and design of the corresponding assays.

The goal of this review is to summarize the latest findings of pathogenetically significant changes in circulating nucleic acids (somatic mutations, non-coding RNAs, and gene methylation patterns) that could serve as candidate biomarkers in liquid biopsy applications. We brief the accumulated data from recent reports on the tumor-associated changes in cfNA applicable as informative tools to diagnose and predict the therapy for CRC and its monitoring. We also discuss the application of state-of-the-art high-throughput methods for the discovery of biomarkers and development of analytical systems.

## 2. Circulating Tumor DNA: Mutations

### 2.1. CtDNA Origin, Amount and Application in CRC

Cancer development and progression are associated with the accumulation of somatic mutations caused by the changes in DNA nucleotide sequence during the life of a cell. CfDNA is released via apoptosis, necrosis, and active secretion from tumor cells into body fluids, such as the blood. The CfDNA pool primarily consists of the germline DNA from normal cells and contains rather minor and highly variable fraction originating from tumor cells that is, circulating tumor DNA (ctDNA) [11]. Up to 1% of the total cfDNA in tumor-bearing patients is ctDNA. The use of ctDNA in clinical tests is determined by the possibility to detect somatic genomic alterations, such as mutations, microsatellite instability, copy number variation (CNV) and the aberrant methylation of DNA. The ctDNA can be recovered from the blood using methods that are routine in clinical labs and can act as a dynamic biomarker for cancer detection and post-treatment follow-up [8,9] (Figure 1).

The early CRC diagnosis and detection of noncancerous adenomas based on the analysis of specific mutations in the circulation are difficult because of the paucity of tumor DNA [12]. Using droplet digital PCR (ddPCR) techniques, Liebs et al. [13] found out that the detection of common *KRAS* and *BRAF* point mutations in plasma was only sporadically successful in the cohort of stage I–III CRC patients; however, 68% of the patients with distant metastasis demonstrated the presence of mutations in cfDNA. Moreover, the levels of ctDNA in CRC increase with the tumor volume enlargement and metastasis development. It was shown that 20% of CRC patients had metastases when diagnosed with 20% of them metastasizing during the follow-up being under systemic therapy. As has been shown recently, metastatic CRC (mCRC) is a complex disease with considerable molecular heterogeneity rather than a single entity. Therefore, the current ctDNA applications are under intensive development aimed at molecular profiling when diagnosing mCRC, selecting the targeted therapy, assessing treatment response/resistance, and post-surgery monitoring of minimal residual disease (MRD) [14].

### 2.2. Development of ctDNA Detection Techniques

#### 2.2.1. PCR-Based Techniques

Generally, a functional assay can detect mutations in ctDNA in two ways. One is the targeted approach that aims to detect previously known genetic mutations, such as specific driver mutations that occur in the tumors with high rates [15]. In this case, quantitative polymerase chain reaction (qPCR) or digital PCR (dPCR) platforms are generally used to detect the pre-selected targets in ctDNA. Beads, Emulsion, Amplification, Magnetics (BEAMing) digital PCR and droplet digital PCR (ddPCR) are highly sensitive, as they detect 0.01% of tumor-associated DNA. The PCR-based methods provide high sensitivity, but their limitation is the low number of targets. The first FDA-approved ctDNA plasma-based genomic Cobas EGFR Mutation Test v2 utilized qPCR technology to reach a detection sensitivity of 0.1–0.8%. Since its approval in 2016, *EGFR* ctDNA testing has proven reliable in clinical settings [16]. Other qPCR-based platforms were created aimed to improve specificity of the assay: peptide nucleic acid (PNA) clamping PCR, amplification refractory mutation system (ARMS), and application of surface-enhanced Raman spectroscopy (SERS) for the analysis of PCR products. According to the VALENTINO study (NCT02476045) for plasma ctDNA analysis, these platforms demonstrate high specificity with variable sensitivity [17]. The use of mass spectrometry allowed the development of the conventional PCR with multiplex detection. Commercial UltraSEEK platform (Agena Bioscience, San Diego, CA, USA) comprise a total of 97 hotspots in *KRAS*, *NRAS*, *PIK3CA*, *BRAF*, and *EGFR.* First, multiplex PCR is used to amplify mutations to further identify them using matrix-assisted laser desorption/ionization time-of-flight mass spectrometry [14].

#### 2.2.2. NGS-Based Techniques

Next-generation sequencing (NGS) provides a parallel detection of multiple mutations with a broad coverage of many genes. The sensitivity and specificity of this method for the detection of mutations display considerable variation (0.1 to 1% depending on the used method and platform) [16]. The measurement of ctDNAs using NGS allows doctors to make cancer diagnostics, select the proper treatment according to prognosis, and monitor the response to treatment [18]. Among the benefits of high-throughput NGS is a concurrent detection of many known and unknown mutations, including single-nucleotide variants (SNVs), fusions, copy number variations (CNVs), tumor mutation burden (TMB), and microsatellite status (MSI). On the other hand, the huge amount of data obtained during high-throughput sequencing requires a detailed and extensive annotation in order to identify the clinically significant alterations in the cancer-associated genes. Note that the proportion of variants of unknown pathogenetic significance and conflicting significance in poorly studied populations reaches up to 30–50%, which prevents the correct diagnosis and, consequently, selection of adequate therapy [19].

The main limiting factors of a widespread use of NGS are the cost of this analysis and a low detection limit of ctDNA assays (the lowest allelic frequency of the target alteration) [20]. However, the cost of sequencing runs is constantly decreasing, while new sequencing platforms have appeared, including Safe-Sequencing System (Safe-SeqS), Tagged-Amplicon deep sequencing (TAm-Seq), and Cancer Personalized Profiling by deep sequencing (CAPP-Seq) with the detection limits as low as 0.01% [20]. If the detection of resistance mutations is the goal of the study, a high sensitivity and a broad coverage is needed, with monitoring that focuses on the specificity of mutations. Large panels of hundreds of cancer genes are needed for the genomic profiling of a tumor followed by tumor fingerprint evaluation, resulting in the development of a smaller panel for the ctDNA-based post-therapy monitoring [14]. Weber et al. [21] tested three NGS platforms, namely, custom SureSelect design (Agilent), QIAact Lung UMI Panel (QIAGEN), and AVENIO Targeted kit (Roche), along with ddPCR and MassARRAY. The authors conclude that a considerable variation in the efficiency of commercially available kits for the sensitivity and specificity of mutation analysis highlights the need for the comprehensive validation of the tests before offering them for routine clinical practice [21]. Examples of the existing NGS panels are listed in Table 1.

#### 2.2.3. Concordance between Different Platforms

Both approaches, targeted and untargeted, have their own advantages and disadvantages and are able to fill different niches in the diagnostic practice. A number of studies compared both commercial and in-house platforms, showing their considerable differences in the input volumes of plasma, DNA isolation and quantification methods, and the total cost per analyzed sample [20,32,33,34]. The study by Holm et al. [22] compared ddPCR, fully automated qRT-PCR–based system Idylla (Biocartis, Mechelen, Belgium), and NGS and demonstrated that Idylla displayed the sensitivity of at least the same level as ddPCR in the detection of the pre-selected *KRAS* mutations in the plasma cfDNA of mCRC subjects. Garcia et al. tested three methods for the detection of RAS mutations in cfDNA from patients with metastatic CRC. They used ddPCR, BEAMing and NGS (targeted SWIFT-56G panel, Swift Biosciences, Ann Arbor, MI, USA). The results suggested the complementarity of methods and variability of results between BEAMing and NGS during the detection of RAS mutations in CRC [23]. An evident advantage of ddPCR as compared with Idylla results from its ability to deliver mutation annotation format (MAF) values and detect a wider range of target mutations. Vessies et al. [32] compared ddPCR, BEAMing, Idylla, and COBAS z480 and found them very different in the number of detected *KRAS* mutations, productivity and cost of analysis of one sample. The authors conclude that the platform selection should be made according to the researcher’s aims and available funds [32].

Commercial amplicon-based Firefly CRC panel (Accu-Kit CRC01, AccuraGen, Shanghai, China), containing a total of 216 hotspots in *KRAS*, *NRAS*, *PIK3CA*, and *BRAF* [30], was compared with UltraSEEK Panel (Agena Bioscience, San Diego, CA, USA), containing a total of 97 hotspots in *KRAS*, *NRAS*, *PIK3CA*, *BRAF*, and *EGFR*. The rate of concordance of these two platforms amounted to more than 98% in the ctDNA from mCRC patients [31]. Kastrisiou et al. [35] developed an original cost-effective NGS gene panel based on the hotspots in six genes: *KRAS*, *NRAS*, *MET*, *BRAF*, *ERBB2*, and *EGFR*. Validation of the panel was made using 68 blood plasma samples from 30 mCRC patients with a diagnosis of the first and second disease progression stages. An overall percent agreement of 86% for *RAS* status was found when results from the developed targeted NGS panel were compared against the plasma samples testing by means of BEAMing ddPCR [35].

Nowadays, various nanomaterials are helpful in improving conventionally used PCR- and NGS-based methods and creating novel platforms for ctDNA detection. However, these variants of ctDNA analysis are at the initial stages and have a lot of technical problems yet to be solved, including stability, sensitivity, and specificity. These inventions include Fe–Au nanoparticle-coupling strategy, electrochemical DNA biosensors, and ddPCR analysis using microfluidic techniques. The mentioned techniques are used in academic research and have to be tested and validated involving different clinical samples [17,36,37].

### 2.3. Genetic Alterations in CRC Used for ctDNA Assays

Several important genetic alterations were reported earlier, such as *KRAS* mutation, which accounts for 44%; *NRAS* mutation, which accounts for 4%; *BRAF* mutation, which accounts for 8.5%; *NTRK1* fusion, which accounts for 0.5%; *ERBB2* amplification, which accounts for 2%: *PIK3CA* hotspot mutation, which accounts for 17%; *ATM* mutation, which accounts for 5%; *MET* amplification, which accounts for 1.7%; *RET* fusion, which accounts for 0.3%; and ALK fusion, which accounts for 0.2% [9]. Many of these mutations not only can serve as a hallmark of CRC but also are relevant to assigning the therapeutic strategies and optimizing regimens.

#### 2.3.1. CtDNA for the Therapeutic Decision

The treatment of CRC has been considerably improved recently thanks to the use of new active agents. Chemotherapy is still the main treatment option for mCRC subjects; however, other approaches are used in combination with chemotherapy or as the options for later stages. They include the antibodies to *VEGF* or *EGFR*, multikinase inhibitors, and immune checkpoint inhibitors [38]. A number of ongoing trials are aimed at assessment of the possibility of using ctDNA detection in order to make valid treatment decisions, such as DYNAMIC, GALAXY, CIRCULATE-Idea, ALTAIR, TRACC, and others [39,40].

Immunotherapy gives another new option in cancer treatment, which is more specific compared with chemotherapy and radiotherapy, and we demonstrated its efficiency for CRC patients [14]. The somatic mutations detectable in cfDNA could be instrumental in the rational assignment of immunotherapy and assessment of its effectiveness. For CRC, the total number of somatic mutations, also known as tumor mutation burden (TMB), could be one of the parameters to assess the response to immunotherapy [41,42] A high level of TMB (≥20 mutations/Mb) was found to be associated with microsatellite instability (MSI) or mutations in the two DNA polymerases (POLD and POLE) [42,43]. In a phase II trial, ctDNA was shown to be helpful in assessing the variant allele frequencies (VAFs) as a predictive biomarker for the immunotherapy of patients with solid tumors treated with an anti-PD1 agent. High pretreatment VAFs were associated with a poor overall survival, while the on-treatment decrease in VAF correlated with longer progression-free survival and overall survival [44].

The selection of adequate therapy with antibodies to EGFR was found to be dependent on the hotspot *RAS* mutations, predicting the resistance to the therapy [45]. The next most relevant is *BRAFV600E* mutation because of its importance in prescribing/following encorafenib and cetuximab combination therapy. Mutations in *MLH1*, *MSH2*, *MSH6*, and *PMS2*, causing microsatellite instability, were shown to be associated with sensitivity to pembrolizumab and nivolumab. Among the other gene alterations potentially influencing the therapeutic decisions are *NTRK* fusions (after treatment with NTRK inhibitor, like entrectinib, for metastatic CRC) or *ERBB2* amplification (possibly related to dual anti-HER-2 blockade) [45]. Mutations in *ERBB2, MAP2K1,* and *NF1* and rearrangements of *FGFR2*, *FGFR3*, *ALK*, *ROS1*, *NTRK1*, and *RET* have recently emerged as novel biomarkers of the response to treatment with anti-*EGFR* monoclonal antibodies [46].

It should be noted that the clinical evidence of the already known genetic alterations for prognosis and therapeutic response prediction in cancer is different. In this regard, the ESCAT (European Scale for Clinical Actionability of molecular Targets) classification has been developed, which determines the usefulness of therapeutic targets for the selection of targeted therapy for metastatic cancer patients, distributing them into six levels. According to the ESCAT, the most recommended for testing in mCRC are RAS mutations and MSI/dMMR (tier IA), as well as BRAF mutations (IB tier), as far as they are relevant for the selection of the first-line therapy in mCRC according to the results of clinical trials. DPD deficiency, NTRK fusion, and HER2/ERBB2 amplification belong to the IIIA and IIIB tiers, respectively; they can influence the treatment plan after at least first-line progression. ALK and ROS1 gene fusions. Meanwhile, mutations of PIK3CA and HER2-activating mutations are IVB tier, which means that they are not yet recommended outside clinical trials and require further verification and testing [47].

Two comprehensive genome profiling liquid biopsy tests for CRC—FoundationOne Liquid CDx and Guardant360 CDx—have been recently approved by FDA for detection of the genomic changes in the cancer-associated genes. Both panels were recommended as matching companion diagnostic tests for treatment selection following professional guidelines. These tests were also approved by the FDA as complementary diagnostics for a number of targeted molecular therapeutic strategies aimed at NSCLC, breast, prostate, and ovarian cancers [48]. According to the recent report, the Guardant360 CDx platform, when used correctly, could identify 28 of 29 (96%) of pretreatment plasma samples of CRC patients as bearing an amplification of *ERBB2*, thus predicting positive response to HER2-targeted therapy [48].

Describing the predictive markers which have potential to make the selection of cancer treatment more precise, one should mention inhibitor of apoptosis proteins (IAPs). The dysregulation of apoptosis due to the overexpression of IAPs plays a critical role in the CRC development, thus representing the therapeutic target [49]. A number of the Smac-mimetics have been designated in order to antagonize IAPs and restore the ability of cancer cells to die via apoptosis. Clinical trials demonstrated their positive anticancer effect in a number of cancer types including CRC especially when they have been used in combination with chemo- or immunotherapy (NCT02890069, NCT02587962). Future studies aimed to test the predictive values of the increased AIPs either at the mRNA or protein level in combination with the genetic aberrations using liquid biopsy seem to have high potential for the CRC predictive tests improvement.

#### 2.3.2. CtDNA in the Treatment Response Evaluation

Even when the prediction almost ensures a positive response to a targeted therapy, the acquisition and clonal selection of mutations conferring resistance to targeted therapy can lead in some patients to severely reduced response or its absence [46]. According to Yi et al. [50], the number of somatic mutations in ctDNA increased after the therapy and the rates of tracked mutations positively correlated with targeted therapy. The ctDNA could be a valuable tool to assess the CRC molecular evolution caused by the effect of different therapeutic courses [15]. Recent studies suggested that the detection of mutated *RAS* in ctDNA would contribute to improving the clinically-based selection of mCRC patients to be re-challenged with anti-EGFR retreatment in combination with immunotherapy (cetuximab and avelumab). Patients retreated with cetuximab and avelumab in the third line of therapy with wild-type *RAS/BRAF* ctDNA at baseline demonstrated the improved OS and PFS as compared with the patients with mutated ctDNA [51]

So far, there is no method allowing for ctDNA quantification and assessment of the tumor burden that would adequately monitor the treatment response [25]. Most recent studies have used targeted sequencing to measure somatic VAF and track mutations with the highest VAF as a proxy measure for the quantification of ctDNA. Lim et al. [29] in their study of the 93 mCRC patients receiving chemotherapy compared the average of VAF of all mutations in 16 genes (*APC*, *TP53*, *KRAS*, *PIK3CA*, *BRAF*, *EGFR*, *ERBB2*, *ERBB3*, *FGFR1*, *NRAS*, *HRAS*, *IRS1*, *MAP2K1*, *MET*, *PDGFRB*, and *PTEN*) during the course of chemotherapy. According to the results, new mutations appeared at the time of resistance, and their number was associated with the response to treatment, and the average VAF in ctDNA was found to be changed as well.

The commercial Oncomine™ Colon cfDNA Assay (ThermoFisher, Yokohama, Japan) produces 48 amplicons covering 240 key hotspot mutations of 14 genes (*AKT1*, *BRAF*, *CDKN2A*, *CTNNB1*, *EGFR*, *HRAS*, *KRAS*, *NRAS*, *IDH1*, *IDH2*, *PIK3CA*, *TP53*). Using this assay, Manca et al. [27] studied the background VAF as a prognostic marker in the cancer patients with wild-type *RAS* treated with an anti-EGFR–based upfront strategy (FOLFOX/panitumumab) in the VALENTINO study (NCT02476045) and demonstrated that *TP53*, *APC*, *PIK3CA*, *SMAD4*, and *FBXW7* were among the key genes contributing to the VAF value [27]. In another study, Osumi et al. [28] also used Oncomine assay for the plasma cfDNA detection in the 101 patients with mCRC who received chemotherapy. The authors demonstrated clinical utility of the targeted NGS cfDNA panel for 14 genes as far as they found an association between ctDNA and clinical factors in the mCRC patients [28]. Subki et al. analyzed a combination of protein CEA and CA19-9 with KRAS mutations in cfDNA of blood plasma from 183 patients with rectal cancer. CA19-9 levels and KRAS mutational status demonstrated a strong association. The increase in protein markers in the presence of a positive KRAS status was associated with a poor prognosis in patients with RC [26].

Wei et al. [52] applied targeted NGS covering exons of 170 genes to assess the *HER2* copy number variations in the subjects with *HER2*^+^ mCRC and found that the changes in the ctDNA burden in plasma were consistent with imaging evaluation during the treatment course, thereby demonstrating that ctDNA gives important information about the response to the treatment [52].

To sum up, the value of the circulating DNA mutations as the markers for therapy prediction in CRC has been studied using various advanced techniques for ctDNA testing, including the detection of the pre-determined driver mutations that occur in the tumor at a high rate, as well as the high-throughput sequencing, allowing for comprehensive analysis of ctDNA. Both approaches have their advantages and limitations, making them more or less applicable to CRC diagnosis, treatment prediction/prognosis, and detection of recurrence. Because of low ctDNA abundance in advanced adenoma and early CRC, the developed tests generally show more promise as the tools for treatment decision making in metastatic CRC, including the information about the combined use of conventional and targeted drugs, such as antibodies against *VEGF* or *EGFR*, multikinase inhibitors, and immune checkpoint inhibitors. Serial ctDNA liquid biopsy during the post-treatment follow-up is also highly instrumental in the early detection of MRD, keeping track of tumor evolution, and the emergence of resistant mutations, which is necessary for the timely and adequate assignment of therapies.

## 3. Aberrant Methylation of Circulating cfDNA

### 3.1. Aberrant DNA Methylation in CRC

Considering the significant role of epigenetic alterations in the colorectal cancer initiation and progression, extensive research has been made to identify the new epigenetic features of CRC that could act as biomarkers [53]. One of the main epigenetic marks is the methylation of cytosines in DNA, which can become deregulated during the genome reprogramming associated with cancer and leading to two major epigenetic abnormalities—DNA hypermethylation and hypomethylation. The hypermethylation of the CpG regions in gene promoters may repress individual tumor suppressor genes, whereas the global hypomethylation rather contributes to tumorigenesis by activating oncogenes and promoting genomic instability. An altered DNA methylation pattern is common for most types of cancer, and it takes place in early cancer development and throughout the disease. This makes the abnormal DNA methylation pattern in blood cfDNA an appropriate marker for a wide range of oncological diseases [54].

The early onset and uniformity of DNA methylation events during cancer progression indicate that these events are more reliable for the early CRC screening tests versus somatic mutations, deletions/insertions, and loss of heterozygosity. The studies have demonstrated the concordance of the alterations in methylation patterns of cfDNA with the DNA from paired tumor samples, making the circulating abnormally methylated DNA a promising cancer biomarker [55,56,57]. Liquid biopsy tests for DNA methylation could be particularly useful when combined with the already established screening approaches and medical imaging to greatly improve the cancer patient outcomes. Their potential areas of application cover the primary diagnosis, selection of therapy, monitoring of the response to treatment, its success, residual disease, and early detection of relapses [54,58] (Figure 1). The study of circulating methylome gives a better insight into the onset of disease and the evolution of cancer phenotypes [16]. Moreover, the possibility of a targeted impact on epigenetic changes determines the new approaches to the treatment of cancer diseases [16,58].

### 3.2. Techniques Used for the Aberrant cfDNA Methylation Analysis

The variety of the corresponding techniques, as well as their advantages and drawbacks, is reviewed elsewhere [59,60,61,62,63]. The methods based on bisulfite conversion are the most commonly used methods for DNA methylation studies. The bisulfite-converted DNA can be analyzed using different platforms. Pyrosequencing and PCR are two major groups of technologies for locus-specific DNA methylation assays. The methylation-specific PCR (MSP) techniques have been widely used for the detection of abnormal DNA methylation, such as quantitative MSP (qMSP), high-resolution melting analysis (MS-HRM), and droplet digital methylation-specific PCR (ddMSP).

Whole-genome bisulfite sequencing (WGBS) gives the most detailed information on DNA methylation profiling and is used for marker selection. The more cost-efficient study of the DNA methylation profile is a reduced-representation bisulfite sequencing (RRBS). This method was designed by combining bisulfite conversion, *Msp*I digestion, and NGS [64]. The amplification of methylated CpG regions and their sequencing (MCTA-seq) represent a highly sensitive method for the detection of hypermethylated sites; its application has allowed for the identification of numerous cfDNA hypermethylation markers useful for the efficacious detection of CRC. The targeted bisulfite sequencing is more appropriate for clinical practice as being scalable and economical, concurrently providing a deeper sequencing coverage. Methylation arrays have been widely used in the search for methylation biomarkers of cancer before the prevalence of NGS [61,62,65].

Methylation-dependent restriction enzymes (methylation-sensitive and methylation-insensitive) used to cleave DNA at specific sites in the nucleotide sequence form the basis for the classical method in methylation studies. This principle forms the background for many array hybridization methods allowing for the detection of 5-methylcytosine (5mC). The advent of ddPCR makes it possible to quantify DNA methylation by using the digestion with methylation-sensitive restriction enzyme (MRE) [66]. In bisulfite DNA sequencing, DNA degradation presents a problem, as well as false positivity; this suggested the development of the new technologies that did not need a bisulfite treatment of DNA. The enrichment-based approaches rely on the use of anti-methylcytosine antibodies or methyl-CpG binding proteins to obtain the methylated regions for further analysis [67]. As a result, an immunoprecipitation-based assay was designed to provide a specific enrichment of methylated cfDNA with subsequent high-throughput sequencing analysis (cfMeDIP-seq). This technique gives the information about the methylation of an approximately 100 bp genomic region without any DNA bisulfite conversion. The methyl-CpG binding domain (MBD) of methyl-CpG binding proteins (MBD2 or MECP2) is also applicable to extract the methylated DNA fragments using magnetic beads. The new DNA assays for quantifying methylation, such as enzyme-based DNA conversion methods (Enzymatic Methyl-seq [67,68,69], in combination with high-throughput sequencing techniques, provide an increased yield of amplifiable DNA as compared with the bisulfite conversion approach, thereby increasing the total number of analyzed CpG sites [69].

Earlier, cfDNA studies have utilized the verification of preselected single-copy genes aberrantly methylated in tumor tissues [70]. Later on, the studies usually combine high-throughput analysis of the DNA from tissues of CRC patients as the first step for marker discovery followed by the validation of the markers in the blood plasma using PCR-based techniques [71,72,73]. However, the strategies utilizing tissue samples for discovering the markers of cfDNA methylation entail a decreased sensitivity of the selected markers that have been tested in the blood serum or plasma. The methylation biomarkers discovered using pooled serum cfDNA as an alternative to tissue samples have been proven to be reliable [65]. Using the MethylationEPIC BeadChip array, Gallardo-Gómez et al. [65] recorded the methylation levels of CpG sites across the genome in the pooled serum cfDNA from advanced adenomas, CRC patients, and noncancer controls. Their results suggest that the assay of 518 differentially methylated positions in the cfDNA distinguishes the cancer-free controls from advanced adenomas or CRC. Heiss et al. [74] used the whole blood for biomarker discovery but emphasized the potential non-specificity of the methylation signature identified in leukocyte DNA for CRC because it merely reflected immune responses.

### 3.3. Methylation Markers in CRC Developed for cfDNA Assays

To date, the development of the CRC assay has been reported as associated with the hypermethylation status of a number of single-copy genes, such as *RARB*, *RASSF1A*, *APC*, *MGMT*, *ITGA4*, *MAP3K14-AS1*, *IKZF1*, *CLDN1*, *MSC*, *INHBA*, *SLC30A10*, *BCAT1*, *SEPT9*, *GRIA4*, *SLC8A1*, *SYN3*, *T-UCRs*, *TMEM240*, *EYA4*, *GRIA4*, and *EHD3* detected in the cfDNA from the blood plasma (Table 2). An alternative strategy was used to consider the hypomethylation of genetic mobile elements (LINE-1, Alu, SINE, and ERV), which were dispersed in the human genome as multiple repeated copies. A repetitive DNA-based assay was proposed to improve the sensitivity as compared with a single-copy gene assay [74,75].

Several individual markers were shown to be very promising for CRC detection, such as *SEPTIN9*, *EHD3*, *TMEM240*, *SMAD3*, and *NTRK3.* The panels of abnormally methylated genes have been constructed to increase the sensitivity of the test, namely, *BCAT1*, *IKZF1*, and *IRF4*; *SEPTIN9* and *HLTF*; *CLDN1*, *INHBA*, and *SLC30A10*; *EYA4*, *GRIA4*, *ITGA4*, and *MAP3K14-AS1*; *DLX5*, *FGF5*, *FOXF1*, *BCAT1*, *COL4A2*, *GRASP*, *IKZF1*, *IRF4*, *SDC2*, *FOXI2*, and *SOX21* (Table 2). It is noteworthy that the marker sets do not overlap. One of the reasons is that the techniques for cfDNA methylation analysis differ between the studies. The above-mentioned PCR-based technologies, BeadChip arrays, and sequencing techniques have been widely used for the discovery of reliable methylation markers and their detection in cfDNA (Table 2).

A recent work of the team led by Lin [70,70,77,84] gives an example of methylation CRC markers designed starting from the in-depth study. They conducted a bioinformatics analysis of data on the DNA methylation profile of a tumor and normal tissue available for research in the Cancer Genome Atlas. Using the TCGA data from colon, rectal, gastric cancer patients and their own results of the Illumina Methylation 450K BeadChip array analysis of a CRC patient tumor and normal tissues, the authors selected *EHD3*, *TMEM240*, and *SMAD3* as promising tumor markers [84]. They validated *EHD3*, *TMEM240*, and *SMAD3* on blood plasma samples from an independent group of CRC patients using the methods most accessible for clinics, namely, methyl-specific/methyl-sensitive PCR (Table 2). The results suggest that *EHD3* is a potentially predictive marker as far as the response rate for chemotherapy was significantly higher in the patients displaying low circulating *EHD3* methylation levels [77]. The hypermethylation in circulating *TMEM240* was found to be valuable for the prognosis and early recurrence in colorectal cancer [70,70]. The circulating hypomethylated *SMAD3* demonstrated potential as an early CRC marker [84].

Barault et al. [78] assessed DNA methylation on a genome-wide scale using Infinium HumanMethylation450 BeadChip arrays to estimate the utility of methylation markers in cfDNA for assay in the monitoring of treatment response rather than a method for early detection. They analyzed a large cohort of CRC cell lines and compared it to the normal mucosa from noncancer patients and blood cells (using the results extracted from databases) [78]. A five-gene signature was defined (*EYA4*, *GRIA4*, *ITGA4*, *MAP3K14-AS1*, and *MSC*) as valuable for the monitoring of treatment response in mCRC. Highly quantitative digital PCR-based assays (methyl-BEAMing) were designed and used to assess the prevalence of five genes in the cfDNA from mCRC patients and healthy donors (Table 2). Assuming that a positive response to only one marker was enough for tracking the tumor burden, they showed the positivity of 87% of the cases. The dynamics of the selected methylation markers correlated with objective tumor response and PFS.

Mitchell et al. [72] applied two approaches to assess genome-wide methylation, namely, SuBLiME and Bisulfite-tagging (Table 2). They developed a panel of markers comprising 11 genes (*COL4A2*, *FGF5*, *FOXI2*, *GRASP*, *IKZF1*, *IRF4*, *DLX5*, *SDC2*, *SOX21*, *FOXF1*, *BCAT1*) displaying a low methylation level in the DNA from the blood of healthy donors, which emerged to be appropriate for evaluation as blood-based diagnostic markers [72,85,86]. Further validation allowed the authors to select a two-marker panel (*BCAT1* and *IKZF1*). Notably, the methylated *IKZF1* and *BCAT1* genes exist as a COLVERA™ liquid biopsy test. It is today available in the United States as a lab-developed test (LDT) for the detection of residual disease and for the monitoring of relapsing CRC [87]. COLVERA™ gives sufficient information about the surgical resection completeness, relapse-free survival and risk of residual disease [87]. Recently, this panel was expanded to three markers (*BCAT1*, *IKZF1*, and *IRF4*) detected by multiplexed real-time PCR and showed efficient detection on cfDNA [77]. The multi-panel test displayed a high sensitivity for CRC and HGD adenomas (71 and 23%, respectively) in the case of a positive result, which included those with ≥ 1 PCR replicates positive for either *IKZF1* or *IRF4,* or at least two replicates positive for *BCAT1,* and a significantly improved specificity (94%) versus any PCR replicate positive.

Zhang et al. [83] analyzed a pre-selected panel of the DNA methylation profiles of 21 genes according to the manufacturer’s instructions (AnchorDx, Guangzhou, China). According to the results, they developed a model comprising circulating markers *C9ORF50*, *TWIST1*, *KCNJ12*, and *ZNF132,* which discriminated the CRC patients from healthy donors with a specificity of 97% and sensitivity of 80%. The authors suggested that this four-marker methylation model provides a new noninvasive choice for the CRC screening [83].

### 3.4. Commercially Available cfDNA Methylation Assays

#### 3.4.1. Methylation Marker-Based Tests for CRC Diagnostics

So far, a few DNA methylation-based markers have been implemented as commercially available assays for CRC diagnostics; *SEPT9* is one on the list. In 2016, Epi proColon was adopted as the first FDA-approved blood-based CRC screening test to detect methylated *SEPT9* (mSEPT9) in cfDNA [88]. Epi proColon 2.0 CE displays improved specificity and sensitivity as compared with the first-generation Epi proColon test. But this test is not recommended by the US Preventive Services Task Force in their guidelines because of low sensitivities for early-stage CRC and advanced adenomas. The meta-analysis by Song et al. showed a better performance of the mSEPT9 assay as compared with the serum protein biomarkers and fecal immunochemical test (FIT) in symptomatic patients [89]. In a meta-analysis comparing CRC patients with healthy donors, the *SEPT9* methylation detection demonstrated 74% specificity and 96% sensitivity [90]. Recently, Loomans-Kropp et al. [81] evaluated the informativeness of the Epi proColon^®^ V2.0 test as a screening tool for early-onset CRC (EOCRC) defined as a CRC diagnosis under the age of 50 years. They showed that the sensitivity and specificity of the mSEPT9 assay detecting EOCRC were 90.8 and 88.9%, respectively, in the discrimination of CRC from healthy subjects; this suggests the test’s applicability for EOCRC detection [81].

Another commercial CRC screening assay (ColoDefense test, VersaBio, Kunshan, China) detects *SEPT9* and *SDC2* methylation. The sensitivities of this kit for the detection of stage I CRC, stage I–IV CRC, and advanced adenomas are 74, 88, and 45%, respectively, with a specificity of 93%. An improved sensitivity for the CRC early stages makes this test useful for early CRC screening, auxiliary diagnosis, and prognosis of postoperative recurrence [80]. Methylated *SDC2* as an individual marker in stool DNA was suggested for CRC diagnostics. The detection of two *SDC2* gene fragments (named *SDC2-A* and *SDC2-B*) improved the sensitivity of the assay. The specificity and sensitivity for cancer detection using two fragments in combination (*SDC2-A*, *SDC2-B)* was 87 and 95%, respectively [91]. Using an additional marker, *SFRP2* MethyLight assay, a new plasma-based technique for CRC early screening (SpecColon test), was recently designed. SpecColon test combines the detection of methylated *SFRP2* and *SDC2* in one qPCR reaction and shows sensitivities of 58 and 76%, respectively, with a specificity of 88% [92]. In the study of Zhao et al. [93] the performance of the SpecColon stool test was assessed for early CRC detection and demonstrated that its sensitivities in the detection of advanced adenomas, early stages (I–II) CRC, and stages I–IV CRC were 54, 89, and 84%, respectively, with a specificity of 93% [93]. The above-mentioned marker pair *IKZF1*, *BCAT1* is also potentially beneficial for primary CRC diagnosis as well as shows a better performance as compared with the Epi proColon^®^ SEPT9 test [87].

#### 3.4.2. Application of Methylation Markers for the Multi-Cancer Tests

Several studies are aimed at construction of the panels of methylated markers able to diagnose several cancer types, including CRC. Whole-genome bisulfite sequencing (WGBS) was used by the GRAIL biotech company for the detection of multi-cancer cfDNA signatures in their Circulating Cell-free Genome Atlas study (CCGA) [94]. The targeted methylation-based assay was improved and validated in the clinical sub-studies, demonstrating a sensitivity of 82% for CRC detection and specificity of 99.5% for 12 cancer types. PanSEER is another promising blood-based cancer screening test utilizing a large panel of markers for ctDNA methylation aimed at the early diagnosis of colorectal, lung, esophageal, stomach, liver cancers [9]. The assay was developed to examine 477 cancer-specific differentially methylated regions (DMRs) indicating cancer specificity and associated with 657 genes. Semi-targeted PCR libraries were created of bisulfite-converted cfDNA and assayed using NGS. At a fixed specificity of 96%, the assay displayed an overall sensitivity of 88% in the post-diagnosis group and 95% in the pre-diagnosis group, together with consistent sensitivity for patients with the disease diagnosed 1 to 4 years later [9]. IvyGene^®^ is another recently commercialized technology (Laboratory for Advanced Medicine) utilizing the methylation signatures (46 markers) and identifying four cancer types according to cfDNA, CRC included. The platform was developed using NGS and bioinformatics analysis. This study aimed at the detection of specific hypermethylated gene targets and the construction of commercial methylation-based biomarker panels [95]. The test is based on targeted PCR and NGS technologies and is able to detect colorectal cancer with a sensitivity of 93% and specificity of 100% [95].

### 3.5. Marker Combination for Development of Low Invasive CRC Test

The CancerSEEK study integrated a panel of genetic mutations (16 genes; 1933 genomic positions total) and eight protein biomarkers from the blood plasma, achieving a good performance in identifying eight common cancer types. The specificity of this multi-cancer screening test for CRC was over 99%, but the sensitivity for CRC was only around 60% [15].

The FDA approved recently the stool DNA test that is available under the commercial name of Cologuard^®^ (Exact Sciences, Madison, WC, USA). It comprises methylated *BMP3* and *NDRG4*, seven *KRAS* mutation sites, and an immunochemical assay for human hemoglobin. Cologuard test detects 92% of CRC and 42% of advanced adenomas with a specificity of 86% [15]. This test is recommended by the US Preventive Services Task Force in their guidelines for the risk screening in asymptomatic individuals aged 50–85 years.

The LUNAR-2 blood test (Guardant Health, Palo Alto, CA, USA) combines the detection of methylation alterations, somatic variants and other epigenomic changes. It is characterized by high sensitivity in detecting CRC in the preliminary study [96]. Guardant Health initiated the prospective, multi-site registrational study (ECLIPSE) for early CRC detection with the LUNAR-2 test. This study has estimated the enrollment of approximately 20,000 subjects aged 46–84 with an average risk of CRC [97].

To sum up, the assessment of aberrantly methylated ctDNA in the plasma/serum has shown a strong potential to become a viable noninvasive alternative and/or accompanying test to the current screening for CRC, a characteristic of which is frequent asymptomatic development, as well as prognosis, prediction, and the treatment follow-up. Nevertheless, our understanding of CRC epigenetics in tissue and blood is far from complete and needs further studies for evaluation. As is known, methylation profile is affected by age, gender, and, for example, smoking; this should be kept in mind when using aberrant methylation patterns in diagnosis. Application of the modern high-performance technologies and the development of advanced techniques are necessary to create the informative tests, which should be further validated by extensive clinical trials.

## 4. Circulating Non-Coding RNAs

### 4.1. Circulating microRNAs in CRC

MicroRNAs represent a group of small non-coding RNA molecules 18–25 nt long, which are the most studied class of non-coding RNAs (ncRNAs). Altered miRNA expression patterns have been observed in almost all cancer types, in that miRNAs can either promote or suppress tumor development [98,99,100,101]. Tumor-specific miRNAs have been recorded in the blood plasma/serum of patients in detectable concentrations and have been found to be sufficiently stable during the preparation of clinical samples and their storage, thus attracting particular interest as cancer markers [102,103,104]. Circulating tumor-specific miRNAs have been comprehensively examined as the tumor markers; however, multiple utilization challenges remain [105]. The biases can be related to the procedures of sampling, storage, and isolation as well as insufficiently standardized miRNA quantification, miRNA selection, sample type, normalization techniques, and patient group selection [104,106,107].

#### 4.1.1. MiRNA Quantification Techniques

The currently available methods for miRNA quantification differ in the throughputs, cost, dynamic range, and sensitivity. The quantitative reverse-transcription PCR (qRT-PCR) is the most popular technique for the pre-selected miRNA quantification thanks to its applicability in routine clinics. Stem-loop RT-PCR is applied to detect mature miRNAs and can distinguish the miRNA species differing in only a single nucleotide. Other approaches include ddPCR, bead, or particle detection [108].

Microarray and NGS are commonly used to measure the overall miRNA amounts. The main advantage of microarrays is that they measure the expression levels of hundreds of miRNA genes jointly [108]; however, they are inapplicable for absolute quantification and display lower sensitivity and specificity as compared with the other methods. NGS offers a higher sensitivity and a wider dynamic range relative to microarray profiling and is appropriate for constructing a profile of all small RNAs present in a sample, miRNAs and other ncRNAs included. However, analysis of these data needs rather complex computational and bioinformatics assistance; moreover, every sequencing platform may produce its own specific artifacts [104].

The universal reference miRNA definition for the normalization of miRNA expression data is one of the challenges to be solved. The adjustment of measurements to the mean of miRNAs expression in samples when analyzing microarray or sequencing data is used in many normalization techniques (for example, trimmed mean of M-values, TMM) [108]. However, it is not applicable to a small panel of measured miRNAs. A number of studies attempted to find optimal reference miRNAs for analyzing qPCR data. Niu et al. [109] used ddPCR to analyze the blood serum of subjects with colorectal, lung, and breast cancers in comparison with healthy donors. The authors inferred that hsa-miR-106b-5p, hsa-miR-93-5p, and hsa-miR-25-3p expression levels were stable in the groups and are applicable for the qPCR data normalization in CRC [109]. These reference miRNAs are not universal in different studies [110,111]. A pairwise normalization was found to be a useful alternative approach giving the advantage of possible enhancement of the effect of both individual miRNAs, so that there was no need for additional normalization [112].

#### 4.1.2. Search for miRNA Markers for CRC Diagnostics

Many recent studies start from a massive miRNA expression analysis using NGS or microarray followed by the validation of candidate marker miRNAs using an amplification-based assay. According to Dansero et al. [106], approximately half of the studies used results of their own preliminary studies aimed for the miRNA markers selection; another half relied on the reported results in making their choice of miRNAs [106]. The authors conducted a meta-analysis for hsa-miR-21 only as far as it was the most frequently reported miRNA over the 44 studies that were included. The deregulation of circulating hsa-miR-21 allowed CRC diagnosis with 77% sensitivity and 82% specificity; notably, hsa-miR-21 was stable and always upregulated in CRC. The other hsa-miRs, namely, hsa-miR-31, hsa-miR-15b, has-miR-20a, hsa-miR-210, hsa-miR-25, hsa-miR-139-3p, hsa-miR-29b, hsa-miR-18a, hsa-miR-22, hsa-miR-17, and hsa-miR-29a, were studied two or three times [106]. This result was supported by another systematic analysis by Fathi et al. [113], which aimed to assess the individual efficacies of hsa-miR-21, hsa-miR-29a, and hsa-miR-92a. The overall pooled results for miR-21 in CRC diagnosis sensitivity, specificity, AUC (area under ROC curve), PLR, and NLR were 78%, 91%, 0.95, 8.12, and 0.17, respectively. However, the clinical application of hsa-miR-21 for diagnosing requires further studies and large-scale analysis to upgrade the diagnostic accuracy; it is necessary to standardize the sample processing procedures and techniques aiming to considerably decrease potential variation [113]. In addition, hsa-miR-21-5p expression is related to the CRC recurrence and progression after surgery [114].

Examples of the recently described putative miRNA markers for CRC diagnosis are listed in Table 3. Candidate panels have been constructed from two up to dozens of hsa-miRs (Table 3). These panels aimed to improve CRC diagnosis, prognosis, and prediction of therapy [107] (Figure 1).

Radwan et al. [131] evaluated the capabilities of preselected plasma hsa-miR-211 and hsa-miR-25 for CRC detection using qRT-PCR. ROC analysis has shown the reliability of hsa-miR-211 and hsa-miR-25 for a significant discrimination between CRC subjects and healthy individuals. In addition, the plasma displayed a positive correlation with lymph node metastasis [131]. Orosz et al. used qRT-PCR to find the circulating hsa-miRs differentially expressed in rectal (RC) and colonic cancers in the blood [154]. CRC and RC patients demonstrated the decreased hsa-miR-155, hsa-miR-34a, and hsa-miR-29a when compared with the group of healthy donors. The patients with RC displayed a higher hsa-miR-221 expression level as compared with controls along with higher hsa-miR-155, hsa-miR-21, hsa-miR-221 levels as compared with CRC patients (5S rRNA, *U6sn* RNA reference) [154].

According to recent reports, the high-throughput analysis of miRNAs as the first step in marker discovery has rapidly evolved [108]. Massive sequencing and validation with qRT-PCR allowed Gmerek et al. [141] to find a set of eight circulating hsa-miRs (hsa-miR-21, hsa-miR-195, hsa-miR-17, hsa-miR-20a, hsa-miR-145, hsa-miR-224, hsa-miR-139, and hsa-miR-32), which were differentially regulated in CRC. Note the absence of any overlap of the miRNAs regulated in tissue and serum; this suggests that the marker hsa-miRs may be of a nontumor origin; however, their aberrant expression reflects cancer development [141].

Wang et al. [155] used miRNAme microarray (miRNA UniTag™) analysis in order to compare colorectal patients with healthy donors and found that 39 hsa-miRs were upregulated and 48 hsa-miRs were downregulated. The results suggest the utility of hsa-miR-31, hsa-miR-141, hsa-miR-224-3p, hsa-miR-576-5p, and hsa-miR-4669 (miRNA-16-based normalization) expression profiles for CRC diagnosis [155]. Using GeneChip^®^ miRNA 3.0 Array (Affymetrix), Nagy et al. [103] estimated the expression profile in matched plasma and tissue samples from patients with CRC, two types of adenoma, and healthy subjects. Hsa-miR-149, hsa-miR-3196, and hsa-miR-4687 expression levels in the blood plasma allowed the discrimination of cancer patients from subjects with adenoma. Circulating hsa-miR-612, hsa-miR-1296, hsa-miR-933, hsa-miR-937, and hsa-miR-1207 expression levels were decreased in CRC versus normal plasma samples [103].

The EXIQON miRCURY LNA-based PCR platform (Qiagen, Hilden, Germany) was used to analyze the expression of 179 hsa-miRs in the serum samples of eight CRC subjects and ten controls [156]. Machine learning was used to develop a model for cancer risk prediction. Since the small sample size did not allow separate training and test samples, this problem was compensated by cross-validation. A panel of 29 hsa-miRs upregulated in CRC was obtained; these miRNAs were regularly observable in the examined CRC samples. Repeated analysis of the publicly available hsa-miR profiles of CRC tumors or CRC exosomes demonstrated that two of the selected 29 hsa-miRs were upregulated in all datasets, hsa-miR-34a and hsa-miR-25-3p included [156].

Silva et al. [129] selected a four-marker based signature comprising hsa-let-7e-5p, hsa-miR-106a-5p, hsa-miR-28-3p, hsa-miR-542-5p using a TaqMan low-density array (Applied Biosystems, Waltham, MA, USA). This model was applied to the independent published datasets and demonstrates a good discrimination in five of the eight sets used (AUC = 0.82) [129].

A serum four microRNA-based (hsa-miR-5100, hsa-miR-1343-3p, hsa-miR-1290, hsa-miR-4787-3p) diagnostic model [111] was designed utilizing four microarray datasets with a standardized platform (3D-Gene^®^ Human miRNA Oligo Chip, Toray Industries, Tokyo, Japan). A comparison with the available NGS-based tests showed a superior performance of the developed multi-cancer test in detecting 12 cancers (lung, colorectal, gastric, esophageal, biliary tract, bladder, glioma, liver, pancreatic, prostate, ovarian cancers, and sarcoma) in the case-control validation cohort (3792 serum samples). This diagnostic model allows discrimination between patients with colorectal cancer and healthy subjects with 86% sensitivity and 92% specificity [111].

#### 4.1.3. MiRNA Markers for CRC Prognosis and Therapy Selection

As has been found, several circulating miRNAs are associated with the drug response observed in CRC subjects. In the last 10 years, many studies of CRC focused on the miRNA differential expression in response to treatment [157]. Recently, serum samples of 95 metastatic CRC subjects have been analyzed aiming to assess the expression of 84 pre-selected tumor-related miRNAs using a NucleoSpin miRNA kit (Machnery-Nagel, Düren, Germany) followed by machine learning to identify the miRNA signatures putatively related to the response of mCRC subjects to irinotecan therapy. Several 20 miRNAs with the most pronounced differential expression in the samples have been earlier associated with CRC [158].

Toledano-Fonseca et al. [115] examined the profiles of circulating miRNAs and constructed the predictive models for the metastatic CRC patients for the response to therapy and their survival. The mCRC patients were involved in a clinical phase II trial before the treatment according to FOLFIRI scheme plus aflibercept; in particular, the expression levels of 754 circulating miRNAs were assessed with qPCR in a TaqMan OpenArray Human Advanced microRNA Panel (Applied Biosystems). A total of 47 circulating miRNAs distinguished between the CRC patients who demonstrated the response to the treatment from those who did not. A number of assayed miRNAs had a predictive potential and, correspondingly, were used in the predictive models for the response to therapy, disease progression, and survival of the patients treated according to the FOLFIRI scheme plus aflibercept. It is noteworthy that the joint use of the levels of circulating miR-33b-5p with the protein marker VEGF-A helped clinicians choose metastatic CRC patients for FOLFIRI plus aflibercept treatment [115].

Hong et al. [140] used NGS for the detection of differentially expressed hsa-miRs in UQCRB-expressing cell lines. Analysis of the sequencing results suggested that six hsa-miRs (hsa-miR-4485, hsa-miR-4745-5p, hsa-miR-1908-3p, hsa-miR-12k26-3p, hsa-miR-4435, and hsa-miR-21-3p) were significant; hsa-miR-4435 was selected the final candidate for validation as a potential marker in the detection of colorectal cancer [140].

Numerous studies suggest that circulating miRNA expression levels are the valid markers of CRC initiation, progression, and response to different therapeutic strategies. A number of recent clinical trials on the circulating miRNA markers in CRC are ongoing or have been finished, for example, “Predictive and Prognostic Value of Inflammatory Markers and microRNA in Stage IV Colorectal Cancer”, NCT04149613; “Contents of Circulating Extracellular Vesicles: Biomarkers in Colorectal Cancer Patients (ExoColon)”, NCT04523389; and “Project CADENCE (CAncer Detected Early caN be CurEd) (CADENCE)”, NCT05633342. Thus, the identification of predictive and prognostic panels of circulating miRNAs will most likely impact the clinical practice in CRC patients in the near future.

#### 4.1.4. Circulating Exosomal miRNAs

As is mentioned above, exosomes are among the key players in the tumor metastasis; are abundant in biological fluids; and contribute to stabilization of the biomarkers they harbor, including non-coding RNAs. An ever-increasing attention to exosomes as a source of cancer markers is explainable because they are released at a high concentration from the transformed cells, carry important biological information on their membranes and within their lumens, and preserve this information [159].

Exosomal miRNAs are widely available and highly specific to CRC; therefore, some of these hsa-miRs, such as hsa-let-7a, hsa-miR-1229, hsa-miR-1246, hsa-miR-1229, hsa-miR-150, hsa-miR-21, hsa-miR-223, hsa-miR-23a, and hsa-miR-125a-3p, were proposed to be valuable for diagnosing early and advanced CRC [160,161]. According to Wei et al. [138], the level of an exosomal hsa-miR-193a-5p was found decreased in subjects with CRC. Furthermore, RNA sequence data analysis along with the further validation using qPCR demonstrated that the levels of exosomal hsa-miR-99b-5p and hsa-miR-150-5p were considerably decreased at early CRC stages as compared with healthy subjects [143]. Circulating exosomal miRNAs are able to discriminate patients with metastatic cancer from the subjects without metastases [162]. For example, the expression of serum exosomal hsa-miR-548c-5p is lower in mCRC subjects as compared with the patients without metastasizing [144]. A recent review [161] focuses on circulating exosomal miRNAs in CRC subjects as well as their contribution to CRC progression and therapy. The correlation between exosomes and chemoresistance in CRC patients has been estimated, which suggests that exosomes can play a biological role in the development of treatment response. Recently, Han et al. [139] used hsa-miR microarray analysis with subsequent qPCR verification to identify a combination of circulating exosomal miRNAs, comprising hsa-miR-100, hsa-miR-92a, hsa-miR-16, hsa-miR-30e, hsa-miR-144-5p, and hsa-let-7. This panel can significantly discriminate between chemoresistant CRC subjects and the chemosensitive ones after an oxaliplatin-based chemotherapy in a statistically significant manner [139]. Earlier, an upregulation of hsa-miR-96-5p, hsa-miR-1229-5p, hsa-miR-21-5p, and hsa-miR-1246 in the CRC serum exosomes was observed in 5-FU CRC resistant patients as compared with chemosensitive controls [145].

Despite a large clinical potential, the lack of reliable markers, effective isolation, and sensitive analytical technologies interferes with the clinical translation of exosomes [159]. Any exosome biomarker tests approved by the FDA for cancer diagnosis are currently absent, even though the urine extracellular vesicle test for assessment of the risk of advanced prostate cancer received an FDA breakthrough device designation as an important diagnostic technology [163]. The reported variability of miRNAs is most likely associated with the insufficient standardization of samples as well as different therapeutic strategies selected for individual patients (sample size, clinical, and pathological stages), sampling, sample processing, and different techniques used to assay exosomal hsa-miRs [161,164,165]. As we discuss below, circulating extracellular vesicles and exosomes attract much interest as the source of other tumor-associated ncRNAs. However, there is a pronounced difference in the used sample processing protocols and methods of exosome isolation and the detection of ncRNAs along with the heterogeneity of extracellular components [164,165]. Therefore, the inter-lab collaboration is necessary to standardize and bring extracellular vesicle-based ncRNA markers into clinical practice.

### 4.2. Circulating PIWI-Interacting and Small Nucleolar RNAs

PIWI-interacting RNAs (piRNAs) represent a group of small non-coding RNA molecules, which are approximately 24–31 nt long. These RNAs are among the most important players in the gene expression regulation during normal biological and pathological processes [166,167]. Similar to miRNAs, piRNAs are involved in the post-transcriptional regulation in the cytoplasm [166,167,168]. Currently, the research into piRNAs in CRC is an emerging field. The diagnostic significance has been recently evaluated for piRNA-5937, piRNA-28876, piRNA-020619, piRNA-020450, piRNA-54265, and piRNA-823 (Table 3).

Small RNA sequencing was used for the screening of the piRNA expression profile in the serum samples of seven CRC patients and seven healthy controls [151]. The differentially expressed piRNAs were then assayed in a training sample of 140 CRC subjects and 140 healthy controls by RT-PCR. A two-piRNA panel (piRNA-020619 and piRNA-020450) detected small-size and early-stage CRC cases with high effectivity (AUC was more than 0.8) [151]. Sabbah et al. [149] observed an upregulated piRNA-823 expression in CRC serum and tissues, suggesting its utility as a CRC diagnostic and noninvasive biomarker. The ROC curve test showed a sensitivity of 83% and a specificity of 89% with an area under the curve of 0.93 for the ability of piRNA-823 to diagnose the cases with colorectal carcinoma [149].

The association of piRNA expression level and prognosis potential in CRC was demonstrated by Mai et al. [150,153], who identified the presence of piRNA-54265 in the human serum using a stem-loop RT-qPCR. From the examined 20 piRNAs, piRNA-54265 was upregulated in subjects with cancer as compared with non-tumor tissues, and its increased expression levels in the blood serum correlated with a poor survival rate in a statistically significant manner [150,153]. Note that a recent report suggests that piRNA-54265, detected in the serum, is likely a full-length small nucleolar RNA, SNORD57 [169]. To clarify this important issue, Mai et al. [170] further verified the identity of piRNA-54265 in human serum samples and confirmed their earlier findings.

Small nucleolar RNAs (snoRNAs) are the ncRNAs which are about 60–300 nt long. SnoRNAs are grouped according to their structural peculiarities: H/ACA box snoRNAs and C/D box snoRNAs [171,172]. They are engaged in forming small nucleolar ribonucleoproteins (SNORNPs); thus, they play critical roles in the stabilization, modification, and maturation of ribosomal RNAs (rRNAs). The aberrant expression of snoRNAs was shown to be associated with the development of cancers; however, the diagnostic and prognostic potentials of snoRNAs in the blood of CRC subjects remain vague. Using qRT-PCR, Liu et al. [173] studied the clinical potential of SNORD1C detection [173] in the serum of CRC subjects. Circulating levels of SNORD1C were likely to correlate with the prognosis and poor outcomes in colorectal cancer.

### 4.3. Circulating Long Non-Coding RNAs

Long non-coding RNAs (lncRNAs), which represent a group of non-coding RNAs (200 nt to 10 kb long), attract ever-increasing interest because of their frequently altered expression in different diseases, including various cancer types [174,175]. These molecules are important players in gene expression regulation, alternative splicing, the localization and activity of proteins, and arrangement of cell substructures and protein complexes [176]. The lncRNAs regulate gene transcription via different mechanisms, which are described elsewhere [176,177]. Changes of their expression contribute to all stages of tumor development, including the initial transformation of cells, their proliferation, the migration of cancer cells, and the processes of angiogenesis, invasion, and metastasizing [177]. LncRNAs are able to influence a number of carcinogenic signaling pathways, as mTOR, TP53, PI3K/Akt, EGFR, WNT/β-catenin, and NOTCH [36]. Certain long non-coding RNAs (for example, CCAT1, CRNDE, and CRCAL1-4) are promising candidates showing an altered expression in adenomas, which suggests their potential utility as early CRC diagnostic markers [175,178].

Current technical limitations in the extraction and quantification procedures for circulating lncRNAs need to be removed in order to design reliable protocols for their consideration as clinically useful biomarkers. High-throughput sequencing technologies and advanced bioinformatics approaches allowed for a rapid assessment of the lncRNAs, which showed their abundance and variety of functions. Different estimations announce a wide range of lncRNAs transcribed from the human genome (3000 to 50,000). Different lncRNA microarray platforms are commercially available to measure the expression of more than 30,000 lncRNAs without using the complicated bioinformatics methods required for NGS data processing. The amplification-based techniques, such as qRT-PCR or ddPCR, are the most applicable to the clinics because of their cost effectiveness and simple extraction/interpretation of results as compared with the large-scale technologies. There are commercially available platforms, such as a qRT-PCR platform LncProfiler qPCR Array Kit (SBI), which gives the opportunity to measure levels of 90 lncRNAs simultaneously [179].

Along with the changes in the local expression in cancer tissue, lncRNAs have been found in the blood plasma and turned out to retain their stability, being resistant to RNases [175]. Expression levels of the blood plasma/serum BLACAT1, CCAT1, CRNDE, CCAT2, NEAT1, and UCA1 were found to have diagnostic potential in colorectal cancer [175,180]. Several studies have suggested that exosomal forms of lncRNAs (91H, CRNDE-h, UCA1, TUG1, LNCV6_116109, LNCV6_98390, LNCV6_38772, LNCV6_108226, LNCV6_84003, LNCV6_98602) have potential as additional markers for diagnosis, prognosis, and response to therapy of patients with colorectal cancer (Table 4).

Recently, a high expression of the ITGB8-AS1 lncRNA was observed in CRC [181]. The authors demonstrated inhibited cell proliferation and tumor growth in CRC when the expression of ITGB8-AS1 was downregulated, which suggests the involvement of ITGB8-AS1 in carcinogenesis. Notably circulating ITGB8-AS1 emerged to be detectable in the blood of patients with colorectal cancer and displayed a positive correlation with the differentiation and TNM (tumor, nodus, and metastasis) stage [181]. The serum levels of NNT-AS1 lncRNA are considerably increased in the CRC subjects as compared with healthy individuals (*P* < 0.05). Furthermore, the NNT-AS1 levels are significantly decreased in the postsurgery samples as compared with presurgery ones. In addition, NNT-AS1 upregulation is also observed in CRC exosomes, whereas any significant differences between the serum and exosomes are unobservable in NNT-AS1 levels [182].

Circulating EGFR-AS1 lncRNA was shown to be a potential indicator of tumor burden in the CRC patients [183]. The EGFR-AS1 lncRNA is an antisense transcript of EGFR. The expression of plasma EGFR-AS1 at CRC stage III–IV was elevated as compared with CRC stage I–II CRC. Moreover, plasma EGFR-AS1 levels decreased after the surgery of colorectal lesions in CRC subjects [183].

The accumulating data suggest that lncRNA subpopulations are associated with circulating platelets, microparticles, exosomes, and lipoproteins considered to be enriched with tumor-associated lncRNA markers. The lncRNAs in the circulating exosomes are most often analyzed; they require further study from the standpoint that their altered levels have a potential for CRC detection. Hu et al. [202] using human ceRNA (competitive endogenous RNA) array V1.0 (4 × 180 K; Shanghai Biotechnology Corporation, Shanghai, China) and selected a panel of six exosomal lncRNAs circulating in the blood plasma (LNCV6_116109, LNCV6_98390, LNCV6_38772, LNCV_108266, LNCV6_84003, and LNCV6_98602), which are likely potential noninvasive biomarkers for early CRC diagnosis [202]. The distribution pattern of 79 long RNAs was assessed in three types of circulating extracellular vesicles: apoptotic bodies, microvesicles, and exosomes. Note that the total serum RNA displayed a smaller AUC as compared with exosomal RNA in the same samples. The authors reported a diagnostic significance of lncRNA (BCAR4) combined with two mRNAs (KRTAP5-4 and MAGEA3) from serum exosomes in colorectal cancer [203].

A targeted PCR-based approach demonstrates that circulating GAS5 and hsa-miR-221 detected in the blood plasma as well in the exosomes are valuable for the colorectal cancer prognosis. The expression levels of circulating hsa-miR-221 and GAS5 were associated with the TNM stage, metastases, and other clinical characteristics of patients with colorectal cancer [204]. An elevated exosomal lncRNA 91H expression has a higher risk of tumor relapse and metastasis in CRC subjects [205]. This indicates that circulating exosomal CRNDE-h and lncRNA 91H are the promising markers for diagnosing CRC and predicting its outcome. A qRT-PCR analysis of the expression of 17 lncRNAs in the serum exosomes of cancer subjects demonstrated a downregulation of lncRNA UCA1 in exosomes from the serum samples of patients, while lncRNA TUG1 was overexpressed [206]. The combined ROC curve of TUG1:UCA1 demonstrated high effectivity for the discrimination of subjects with cancer from healthy ones.

Recently, Vallejos et al. [207] used the NGS of the plasma exosome transcriptome and identified 445 genes with highly differential expression, which comprised miRNA, mRNA, and lncRNA. This gene signature, referred to as ExoSig445, showed a good performance in fully distinguishing the colon cancer subjects from healthy controls according to the expression levels; thus, gene panels have a potential of promising highly sensitive liquid biopsy tests [207]. Yu et al. [208] reported the serum exosomal lncRNA profiles of CRC patients and healthy subjects using lncRNA microarray and verified them with qPCR using the samples of 203 CRC subjects and 201 healthy controls. The authors show that lncRNAs FOXD2-AS1, NRIR, and XLOC_009459 have considerably elevated levels in the samples of CRC subjects and have diagnostic potential [208].

### 4.4. Circulating Circular RNAs

Circular RNAs (circRNAs), a group of non-coding RNAs, which are about 200–600 nt long, are single-stranded and have a structure of a covalently closed continuous loop lacking a 5′–3′ polarity formed by back splicing. They redirect miRNAs, stabilize the miRNA binding molecules, and act as a scaffold by binding to different regulatory proteins. CircACC1, upregulated in CRC, induces metastasis, proliferation, and angiogenesis [209].

Systematic analysis by Long et al. [210] demonstrates the significance of circRNAs for CRC diagnosis and prognosis [210]. The discrimination power of the circRNAs for cancer patients versus nontumor controls was found to be moderate (the AUC was 0.81). The survival analysis showed that upregulated circRNAs are significantly associated with a poor survival (HR was 2.38). The upregulated circRNAs in colorectal cancer demonstrated increased diagnostic value as compared with downregulated circRNAs. The efficiency of tissue-derived circRNAs in diagnosing CRC is the same as the efficiency of plasma/serum-derived ones (AUC, 0.81 versus 0.82). Xiao et al. [211] evaluated circRNAs for their potential in diagnosis, therapy, and prognosis for CRC by systematically analyzing 236 papers; this allowed the team to identify 217 circRNAs associated with 108 host genes and 145 miRNAs. From the studied 217 circRNAs, 74 are related to CRC diagnosis; 160 are related to treatment; and 51 are related to prognosis. Some of these circRNAs were exosomal circRNAs, with the valuable characteristics as potential biomarkers [211].

Using Arraystar Circular RNA Microarray Version 2.0, it was shown that eight circRNAs (circ_104885, circ_100185, circ_103171, circ_001978, circ_105039, circ_103627, circ_101717, and circ_104192) had elevated levels in colorectal cancer patients as compared with subjects with colorectal adenoma as well as healthy controls. The validation panel selected for the early colorectal cancer screening contained three circRNAs (circ_001978, circ_105039, and circ_103627). The ROC analysis showed a high diagnostic ability of the selected panel in the discrimination of cancer versus adenoma patients (AUC was 0.96) and cancer patients versus healthy controls (AUC was 0.97) [193].

Mohammadi et al. [190] used qRT-PCR to show an upregulation of circ_0006282 from the plasma of CRC subjects versus healthy controls [190]. The CRC subjects after surgery displayed circ_0006282 with a decreased expression restoring to a normal level after surgery. A joint use of circ_0006282, CEA, and CA199 elevated the diagnostic sensitivity for CRC diagnosis (78.8%) versus a sensitivity of 48.3 and 29%, respectively, which were recorded for CEA and CA199 alone.

The significance of exosomal circRNA forms has been studied as well. Zheng et al. [191] found that exosomal circLPAR1 considerably decreased in the course of CRC progression but was restored after surgery. Exosomal circLPAR1 emerged to be specific for CRC diagnosis and contributed to a better diagnostic performance, which is suggested by an AUC value of 0.88; this was confirmed by the analysis of its performance together with CEA and CA19-9, which are widely used clinical biomarkers [191]. Exosomal circ_0004771 is considerably overexpressed in the CRC subjects and decreased in the serum of CRC subjects after surgery, which demonstrates a diagnostic potential of circulating circ_0004771 for the early stage of colorectal cancer [201].

The lncRNAs and circRNAs have attracted wide attention in cancer liquid biopsy and therapy because of their important regulatory functions. These ncRNAs are involved in the CRC development and progression via regulating proliferation, angiogenesis, immune evasion, metastasis, and therapeutic resistance. Despite a fast increase in our understanding of the lncRNA functional role, many questions and challenges are debatable. Today, there is only a single FDA-approved lncRNA (PCA3)-based test that has reached routine clinical use and been included in the European Association of Urology Guidelines for Prostate Cancer 2019. All documents can be accessed on the EAU website: http://uroweb.org/guideline/prostate-cancer/, 19 March 2019. For men at an increased risk of prostate cancer and previously negative biopsies, the Progensa-PCA3 test is recommended to make the decision about the second biopsy. Results of the ongoing genomic and transcriptomic studies propose that lncRNAs and circRNAs play diverse biological roles; therefore, intensive studies are necessary to discover many more ncRNAs with annotated functions. The comprehensive knowledge about the expression and function of these gene regulation players should enhance the development of novel therapeutics to target gene expression for the treatment of CRC.

## 5. Conclusions

Analysis of the reviewed literature suggests that a large number of studies support the concept that the circulating nucleic acid–based markers are extremely promising for the development of liquid biopsy tests for CRC. Somatic mutations in the blood plasma and serum ctDNA have emerged to be less applicable to early CRC diagnosis but valuable for the prognosis, treatment prediction, early recurrence detection, post-treatment supervision of the tumor evolution, and emergence of resistance mutations. The changes in ctDNA methylation are applicable as sensitive and specific markers for early CRC diagnosis, prognosis, prediction, and tracking of the response to treatment. The plasma/serum or exosome-derived non-coding RNAs, including miRNAs, other types of small ncRNAs, and long ncRNAs, both linear and circular, promise to become the potent sensitive and specific CRC markers detectable with liquid biopsy tests.

The worldwide laboratory studies utilizing high-throughput techniques have allowed for the intensive discovery of CRC markers, their selection, and validation; however, their adoption to clinical practice has been rather slow. As we can find from the analysis of the cited reports, the diagnostic significance of the discovered markers is often based on a limited number of patients and controls, and a predictive validation cohort is rarely included. So far, only a few ctDNA-based liquid biopsy tests for CRC detection have entered the market and been approved as complementary diagnostic tests for clinics. The ncRNA-based tests are now intensively developed, and their potential is tested in ongoing clinical trials.

A number of scientific and technical problems are to be solved to make the potential liquid biopsy tests an everyday method for laboratory medicine. However, further investigations are needed to gain insight into the ways/forms of secretion, circulation, and excretion (platelets, microparticles, microvesicles, and nucleic acid/protein complexes) along with the biological role and pathologic significance of the circulating tumor nucleic acids released into the circulation from tumor/normal cells. The ongoing development of high-throughput and reliable techniques in research and clinical labs has to focus on the selection of the valuable biomarkers. To solve these problems, various methods are developed allowing for mathematical processing of the “big” data resulting from multiomics research. To date, AI algorithms make it possible to detect multiple markers characteristic of tumor and healthy cells and to combine circulating DNAs, RNAs, and proteins into panels to design highly effective tests [212]. The use of AI to identify cell-free biomarkers enabled Freenome to develop a multiomics test that combines tumor- and nontumor signals from DNA and protein biomarkers. The test showed high affectivity for the AA and CRC early detection, and it is currently being validated in a large prospective multicenter study named PREEMPT CRC (https://www.freenome.com/clinical-studies/#colorectal, 1 August 2023). Pre-analytical methods and ctDNA and ctRNA assays analysis need standardization, automation, and certification in order to facilitate a rapid and robust detection and quantification. Most of the candidate ctDNA methylation and ncRNA markers have been identified in small retrospective cohorts or case-control studies, and a few are verified in independent studies. In order to become clinically adequate, individual nucleic acid markers, panels, and signatures demand large prospective cohort studies and population screenings. Significant progress in the area of extracellular RNA biomedical research has been attained thanks to the launch of the NIH Common Fund-supported Extracellular RNA Communication Program (ERSP) in 2013. Further financial support of the international inter-lab collaboration, national government, and medical companies are urgently needed to speed up the progress in the field of cancer liquid biopsy development.

## Figures and Tables

**Figure 1 ijms-24-12407-f001:**
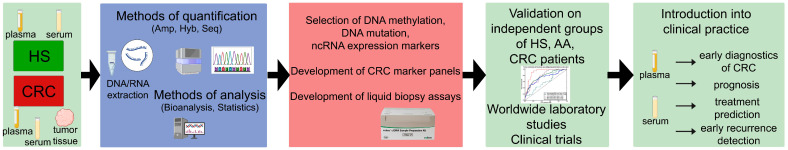
Modern approaches for the development of the liquid biopsy tests for the CRC diagnostics, prognosis, treatment selection, tumor evolution analysis. Abbreviations: HS—healthy subjects, AA—advanced adenoma, CRC—colorectal cancer, Hyb—hybridization, Amp—amplification, Seq—sequencing.

**Table 1 ijms-24-12407-t001:** Mutations in circulating DNA.

Marker(s)	Material	Relevance	Method	Examined Cohort	Reference
*BRAF*, *KRAS*	Plasma	Diagnosis, monitoring	ddPCR	65 patients	[13]
*KRAS*	Plasma	Comparison of methods for measuring KRAS mutations in CRC	ddPCR and NGS	10 mCRC patients	[22]
Genes of *RAS* subfamily	Plasma	Comparison of plasma and tissue RAS analysis in CRC samples	NGS and ddPCR	25 mCRC patients	[23]
*HER2*	Plasma	Early detection of mCRC progression and cetuximab resistance	ddPCR	126 mCRC patients	[24]
		Noninvasive detection of gene mutations in CRC	Immunohistochemistry and fluorescence in situ hybridization	412 CRC patients	[25]
CEA, CA19-9, mutations in circDNA (*KRAS*, *NRAS*, *BRAF*)	Plasma	Prognosis in mCRC	ddPCR	20 mCRC patients	[26]
*TP53*, *APC*, *PIK3CA*, *SMAD4*, *FBXW7*	Plasma	Monitoring after treatment	NGS	135 patients of phase 2 trial	[27]
*KRAS*, *NRAS*, *MET*, *BRAF*, *ERBB2*, *EGFR*	Plasma	Prognosis in mCRC	NGS and BEAMing dPCR	101 mCRC patients, who received chemotherapies	[28]
*APC*, *TP53*, *KRAS*, *PIK3CA*, *BRAF*, *EGFR*, *ERBB2*, *ERBB3*, *FGFR1*, *NRAS*, *HRAS*, *IRS1*, *MAP2K1*, *MET*, *PDGFRB*, *PTEN*	Plasma	Prediction of therapy in mCRC	NGS	93 mCRC patients	[29]
*KRAS*, *NRAS*, *PIK3CA*, *BRAF*	Plasma	Prognosis, prediction of therapy in mCRC	NGS	184 mCRC patients	[30]
*KRAS*, *NRAS*, *PIK3CA*, *BRAF*, *EGFR*	Plasma	Prognosis, prediction of therapy in mCRC	NGS	60 mCRC patients	[31]

Abbreviations: qPCR, quantitative PCR; NGS, next-generation sequencing; and ddPCR, droplet digital PCR.

**Table 2 ijms-24-12407-t002:** Methylation markers in cfDNA.

Marker	Material	Relevance	Method	References
*BCAT1*, *IKZF1*, *IRF4*	Plasma	Screening, diagnostics, and post-treatment monitoring	Multiplex PCR and qMS-PCR	[58,76]
*CLDN1*, *INHBA*, *SLC30A10*	Plasma	Diagnostics	Sequencing (panel, SureSelect^XT^ Methyl-Seq), methyLight assay, and multiplex PCR	[71]
*BCAT1*, *COL4A2*, *DLX5*, *FGF5*, *FOXF1*, *FOXI2*, *GRASP*, *IKZF1*, *IRF4*, *SDC2*, *SOX21*	Plasma	Diagnosis	Genome-wide methylation assessment (SuBLiME and bisulfite tagging)	[72]
*EHD3*	Plasma	Prognosis, prediction of therapy	TaqMan qMS-PCR	[77]
*EYA4*, *GRIA4*, *ITGA4*, *MAP3K14-AS1*, *MSC*	Plasma	Monitoring, prediction of therapy	Methyl-BEAMing assays	[78]
*TMEM240*	Plasma	Diagnosis, prognosis, early recurrence prediction	TaqMan qMS-PCR	[70]
*SEPTIN9*, *SDC2*	Plasma	Screening, diagnostics, monitoring of recurrences	Multiplex qMS-PCR	[79,80,81]
*GRIA4*, *SLC8A1*, *SYN3*	Plasma	Diagnosis, prognosis, screening	MethylBEAMing analyses	[82]
*c9orf50*, *TWIST1*, *KCNJ12*, *ZNF132*		Diagnosis	NGS	[83]

Abbreviations: qMS-PCR, quantitative methyl-specific PCR; MS-PCR, methyl-specific PCR; and NGS, next-generation sequencing.

**Table 3 ijms-24-12407-t003:** Expression of circulating microRNAs and other small RNAs.

Markers/Panel	Material	Relevance	Methods	Clinical Specimens vs. Controls (Predictive Validation Cohort If Applicable)	Reference
	microRNAs				
VEGF-A, hsa-miR-33b-5p	Plasma	Prediction of therapy in mCRC	qRT-PCR	98 patients from phase II trial POLAF + 30 patients as screening cohort (68 patients for validation assay)	[115]
hsa-miR-93-5p	Plasma	Prognosis of early disease recurrence	qRT-PCR	35 CRC patients	[116]
hsa-miR-377-3p, hsa-miR-381-3p	Exosomes in serum	Diagnosis	qRT-PCR	175 CRC patients vs. 172 healthy donors	[117]
hsa-miR-29c, hsa-miR-149	Serum	Diagnosis	qRT-PCR	80 CRC, and 80 colorectal adenoma patients vs. 80 healthy controls	[118]
hsa-miR-92a-3p	Serum and plasma	Diagnosis	qRT-PCR	Summarized data on 874 CRC and 205 adenoma patients	[119]
hsa-miR-762	Serum	Diagnosis	qRT-PCR	20 CRC patients, 20 healthy controls	[120]
hsa-miR-449	Plasma	Diagnosis	qRT-PCR	343 CRC patients vs. 162 healthy controls	[121]
hsa-miR-223	Serum	Diagnosis	qRT-PCR	120 CRC patients and 75 healthy controls	[122]
hsa-miR-1290	Serum	Diagnosis	qRT-PCR	46 PC, 50 CRC, and 50 GC patients vs. 50 healthy individuals	[123]
hsa-miR-19a-3p, hsa-miR-203-3p, hsa-miR-221-3p, hsa-let-7f-5p	Serum	Diagnosis	qRT-PCR	36 CRC patients vs. 30 healthy individuals	[124]
hsa-miR-21	Serum and plasma	Diagnosis	qRT-PCR	Meta-analysis	[125]
hsa-miR-21-5p	Plasma	Prognosis	qRT-PCR	103 CRC after surgical resection	[114]
hsa-miR-15b, hsa-miR-16, hsa-miR-21, hsa-miR-31	Exosomes	Diagnosis	qRT-PCR	Meta-analysis	[126]
hsa-miR-96, hsa-miR-99b	Plasma	Diagnosis and prognosis		110 subjects vs. 20 age- and gender-matched healthy subjects (20 healthy subjects, 41 pCRC, and 49 mCRC patients)	[127]
hsa-miR-21, hsa-miR-210, hsa-miR-203	Plasma	Diagnosis	qRT-PCR	62 stage IV CRC patients vs. 44 healthy subjects	[128]
hsa-miR-28-3p, hsa-let-7e-5p, hsa-miR-106a-5p, hsa-miR-542-5p	Plasma	Diagnosis	qRT-PCR	20 CRC and 21 metastatic CRC patients vs. 68 noncancer subjects (27 healthy controls, 17 individuals with hyperplastic polyps, and 24 with adenoma)	[129]
hsa-miR-618	Serum	Prognosis	qRT-PCR	104 unresectable mCC subjects vs. 90 healthy volunteers	[130]
hsa-miR-211, hsa-miR-25	Plasma	Diagnosis	qRT-PCR	44 CRC patients vs. 40 healthy controls	[131]
hsa-miR-30e-3p, hsa-miR-146a-5p, hsa-miRNA-148a-3p	Serum	Diagnosis	qRT-PCR	137 CRC patients vs. 145 healthy controls (other 80 CRC samples + 88 health controls)	[132]
hsa-miR-21, hsa-miR-92a	Plasma	Diagnosis	qRT-PCR	100 samples: 33 CRC + vs. 37 active UC and 30 IBS subjects vs. 30 healthy controls	[133]
hsa-miR-21, hsa-miR-23a, hsa-miR-27a	Serum	Diagnosis	qRT-PCR	35 CRC patients vs. 35 healthy controls	[134]
hsa-miRNA-585-5p, hsa-miR-15b-5p, hsa-miR-425-3p	Plasma	Diagnosis	qRT-PCR	35 CRC patients vs. 6 tumor-free donors	[135]
hsa-miR-592	Serum	Diagnosis	qRT-PCR	15 CRC patients vs. 15 healthy individuals	[136]
hsa-miR-944	Serum	Diagnosis	qRT-PCR	150 CRC patients and 50 adenomatous polyps patients vs. 100 healthy controls	[137]
hsa-miR-193a-5p	Exosomes in plasma	Diagnosis	qRT-PCR	37 CRC patients, 22 colorectal adenoma, 42 healthy controls	[138]
hsa-miR-100, hsa-miR-92a, hsa-miR-16, miR-30e, hsa-miR-144-5p, hsa-let-7i	Exosomes in plasma	Prediction of therapy	Microarray and qRT-PCR	210 late-stage CRC patients vs. three independent cohorts (47 CRC control patients, 84 responsive patients, and 79 resistant patients (72 responsive patients and 67 resistant patients)	[139]
hsa-miR-4435	Serum	Diagnosing specific stages of CRC	qRT-PCR and RNA sequencing	48 CRC patients at the time of diagnosis	[140]
hsa-miR-375, hsa-miR-486-3p, hsa-miR-486-5p, hsa-miR-1180-3p, hsa-let-7d-5p, hsa-let-7a-5p, hsa-miR-30e-3p, hsa-let-7f-5p	Serum	Diagnosis	qRT-PCR and RNA sequencing	6 CRC patients vs. 6 healthy subjects	[141]
hsa-miR-1290, hsa-miR-320d	Plasma	Diagnosis	qRT-PCR	35 colorectal adenoma and 35 CRC patients vs. 35 healthy controls (80 CRC, 50 adenomas patients, and 30 healthy controls)	[142]
hsa-miR-99b-5p, hsa-miR-150-5p	Exosomes in plasma	Diagnosis	RNA sequencing and qRT-PCR	169 CRC patients vs. 155 healthy donors and 20 benign disease patients	[143]
hsa-miR-548c-5p	Exosomes in serum	Diagnosis in mCRC	qRT-PCR	108 CRC patients	[144]
hsa-miR-21-5p, hsa-miR-1246, hsa-miR-1229-5p, hsa-miR-135b, hsa-miR-425, hsa-miR-96-5p	Exosomes in serum	Prediction of therapy	qRT-PCR	43 CRC patients	[145]
hsa-miR-125b	Exosomes in plasma	Predictive biomarker/therapy monitoring biomarker	qRT-PCR	6 CRC patients vs. 3 healthy volunteers (55 patients with advanced/recurrent CRC)	[146]
miR-612, miR-1296, miR-933, miR-937, miR-1207	Plasma	Diagnosis (screening)	qRT-PCR, microarray	4—normal colonic, 4—tubular adenoma, 4—tubulovillous adenoma, 4—colorectal cancer	[103]
has-miR-210, has-miR-21, has-miR-126	Plasma	Early diagnosis, screening, and predicting prognosis of CRC	qRT-PCR	86 subjects with mass neoplasm according to colonoscopy vs. 101 neoplasm-free controls	[147]
hsa-miR-126	Plasma	Prediction of disease recurrence and response to therapy	qRT-PCR	Patients from a phase II study	[148]
	piwi-RNAs				
piRNA-823	Serum	Diagnosis	qRT-PCR	84 CRC patients vs. 75 healthy controls	[149]
piRNA-54265	Serum	Screening, early detection, and clinical surveillance	qRT-PCR	725 CRC patients, 1303 patients with other types of digestive cancer, and 192 patients with benign colorectal tumors vs. 209 healthy controls	[150]
piRNA-020619, piRNA-020450	Plasma	Diagnosis	RNA sequencing, qRT-PCR	7 CRC patients vs. 7 healthy controls + training sample of 140 CRC subjects vs. 140 healthy controls (180 CRC patients vs. 180 normal controls + 50 lung cancer, 50 breast cancer, and 50 gastric cancer subjects)	[151]
piRNA-5937, piRNA-28876	Plasma	Diagnosis	RNA sequencing and qRT-PCR	403 colon cancer patients vs. 276 healthy donors (179 colon cancer patients + 100 healthy donors)	[152]
piRNA-54265	Serum	Therapeutic target	qRT-PCR	218 CRC patients + 317 additional CRC subjects (215 CRC cases as model set, 102 cases as validation set, and combined samples of both sets)	[153]

Abbreviations: qRT-PCR, quantitative reverse-transcription PCR.

**Table 4 ijms-24-12407-t004:** Circulating long non-coding RNAs and circular RNAs.

Markers	Material	Relevance	Method	Clinical Specimens vs. Controls (Predictive Validation Cohort If Applicable)	Reference
	lncRNAs				
lncRNA ITGB8-AS1	Plasma	Diagnosis and therapeutic target	qRT-PCR	7 CRC patients vs. 7 healthy controls (150 CRC patients)	[181]
lncRNA NNT-AS1		Therapeutic target	qRT-PCR	40 CRC patients before and after surgery, 40 healthy controls	[182]
lncRNA EGFR-AS1	Plasma	Diagnosis and prognosis	qRT-PCR	128 CRC patients vs. 64 age and sex-matched CRC-free healthy individuals	[183]
HOTIIP	Extracellular vesicles in serum	Predictive value/therapeutic target	qRT-PCR, Western blot, and IFA	95 patients with advanced CRC	[184]
B3GALT5-AS1	Serum	Diagnosis	qRT-PCR	45 patients with colorectal polyps, patients, 118 colorectal cancer patients and 88 healthy age-matched controls	[185]
SNHG11	Plasma	Diagnosis and therapy	qRT-PCR	Tumor group *n* = 622 vs. nontumor group *n* = 51 + independent cohort of plasma samples from 90 CRC patients and 44 patients with CPDs vs. 42 age- and sex-matched healthy controls	[186]
DANCR	Serum	Prognosis	qRT-PCR	40 primary CRC patients, 10 recurrent CRC patients, and 40 patients with colorectal polyps vs. 40 healthy controls	[187]
MEG3	Serum	Prognosis	qRT-PCR	126 CRC serum samples vs. 48 healthy controls	[188]
LncRNA-ATB CCAT1	Serum	Diagnosis and therapy	qRT-PCR	74 pretreatment CRC samples vs. 74 controls	[189]
	circRNAs				
circ_0006282	Plasma	Diagnosis	qRT-PCR	100 CRC patients, 25 postoperative CRC patients, 28 colitis patients vs. 108 healthy donors	[190]
circLPAR1	Exosomes in plasma	Diagnosis	qRT-PCR	112 CRC patients and 28 patients with polyps vs. 60 cancer-free controls and patients with other cancer types: 74, gastric carcinoma; 18, breast invasive carcinoma; 24, bladder urothelial carcinoma; 32; 19, kidney renal clear cell carcinoma; and 42, lung adenocarcinoma	[191]
circRHOBTB3	Exosomes in plasma	Prognosis	qRT-PCR	18 CRC patients vs. 12 CRC patients, 21 HCC patients, 32 PAAD patients, and 14 healthy donors	[192]
hsa_circ_001978, hsa_circ_105039, hsa_circ_103627	Plasma	Diagnosis	Microarray and qRT-PCR	100 patients before endoscopic treatment + 100 patients diagnosed with colorectal adenoma vs. 100 healthy donors (20 CRC patients and 20 healthy controls + 80 CRC patients and 80 healthy controls for revalidation)	[193]
circ_PVT1	Plasma	Diagnosis and prognosis	qRT-PCR	148 CRC patients vs. 148 healthy volunteers	[194]
hsa_circ_0005963	Exosomes from serum	Therapeutic potential	qRT-PCR	7 CRC patients	[195]
hsa_circ_0001900, hsa_circ_0001178	Plasma	Diagnosis	qRT-PCR	18 CRC patients vs. 18 healthy donors (80 healthy controls, 30 patients with precancerous lesions, and 102 CRC patients)	[196]
hsa_circ_0004831	Serum	Prognosis	qRT-PCR	81 CRC patients vs. 50 healthy volunteers	[197]
hsa_circ_0002320	Plasma	Diagnosis and prognosis	qRT-PCR	50 patients with CRC before any treatment vs. 100 healthy individuals	[198]
circ-CCDC66	Plasma	Diagnosis	qRT-PCR	Training cohort, 15 CRC patients vs. 15 healthy controls (validation cohort, 30 CRC patients, 46 healthy controls, and 23 disease controls)	[199]
hsa_circ_0035445	Plasma	Diagnosis	qRT-PCR	156 patients with CRC vs. 66 healthy controls	[200]
hsa_circ_0004771	Plasma	Diagnosis	qRT-PCR	170 patients and 45 healthy controls	[201]

Abbreviations: RIP, RNA immunoprecipitation: ChIP, chromatin immunoprecipitation; IFA, immunofluorescence analysis; and RIPA, radioimmunoprecipitation assay.

## Data Availability

The data presented in this study are available upon request from the corresponding author.

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
