# Peer review of "Genomic and Transcriptomic Research in the Discovery and Application of Colorectal Cancer Circulating Markers"

_ijms, 2023, doi:10.3390/ijms241512407_

Round 1

Reviewer 1 Report

The authors provide a comprehensive and highly informative review of the genomic and transcriptomic technological advances in CRC diagnosis, prognosis, prediction, and clinical follow-up. They offer an opinion regarding the benefits, limitations, and challenges of each technology and their application in the detection of various types of nucleic acid molecules and their molecular alterations, including methylation and somatic mutations.

The manuscript is well-written and contains a comprehensive literature review, as well as detailed and valuable information regarding the molecules and assays reviewed.

Suggestion. The manuscript could benefit from the addition of an illustration of the technologies and applications reviewed.

Minor comment: please correct the font differences in lines 135, 181, 644.

Author Response

Dear reviewer! We appreciate greatly your kind attitude and careful analysis of the manuscript. In accordance with your comments, we added the illustration of technologies and applications reviewed, as well as checked up and corrected the font differences.

Reviewer 2 Report

In this manuscript, the authors have summarized the pathogenicity and application of cfDNA in CRC. So far, the authors have been successful in briefing it. To me, this review has been written at its best and the authors didn't leave much space for revision except for some text errors like in Lines 135-136 and Line 181-184. This review is very interesting and might benefit researchers in this direction. Graphical figures representing different types of pathogenicity observed in cfDNA and Different approaches for developing a Biomarker will be really helpful for viewers to understand. Overall, This is a very good review. Good luck!

Author Response

Dear reviewer! We appreciate greatly your kind attitude and very careful analysis of the manuscript. We made significant changes to the text of the article, restructured the text and hope to minimize the plagiarism.

Reviewer 3 Report

In this review paper, the authors investigated the research trends in the discovery and application of circulating DNA and RNA based markers in diagnostic, therapeutic, and preventive strategies of colorectal cancer (CRC). The topic is very interesting. Yet, the plagiarism check of the manuscript shows relatively high similarity ratio (41%) against published literature. So the authors need to carry out substantial major revision to correct this unintentional plagiarism before any peer-review can be performed. The plagiarism check result is attached in this report.

Author Response

Dear reviewer! We appreciate greatly your very kind attitude and careful analysis of the manuscript. In accordance with your comments, we added the illustration of technologies and applications reviewed, as well as checked up the mistakes throughout the text.

Reviewer 4 Report

General 

The screening activity in oncology requires the development of 1) simple sampling methods that are easy to implement and psychologically acceptable to individuals, 2) reliable analytical approaches leading to minimal false positives and false negatives, 3) the ability to integrate the approaches into the care pathway, and 4) cost-effectiveness analysis. Clearly, we are still far from achieving these goals when it comes to the screening of colorectal cancers, and this development concerning circulating cell-free nucleic acids is welcome because it appears that a blood-based test could have public appeal.  According to the authors of this review article, the goal of this review is to summarize the latest findings of pathogenetically significant changes in circulating nucleic acids that could serve as candidate biomarkers in liquid biopsy. 

However the result is a bit far-fetched, mainly because the article is far too long, the structure of the article hasn't been worked on enough, it's largely chaotic. For example, the methods for detecting mutations are discussed beginning at line 193. Based on the division into three main chapters (circulating tumor DNA: mutation, Aberrant methylation of circulating DNA, circulating noncoding RNAS), it is clear that the same organization should apply to each of them. For example: methods of detection, implementation of the methods in the context of cancer screening, including discussion of cost-effectiveness, and finally the use of the methods in the care pathway of patients (evaluation of resistance to treatments for example or prediction of vulnerabilities). In addition the authors should more clearly distinguish between approaches that are robust and suitable for use in clinical practice from those that are still in their infancy.

Specific

  • A graphical abstract is needed that would clarify the state of the art in this domain
  • Citations of references is sometimes misleading and should be carefully revised: see for example ref 40 of table 2
  • The literature should be discussed and/or updated:
    • What about the artificial intelligence machine learning classifier test ( see Freenome)  designed to identify patterns of cell-free biomarkers in the blood for the early detection of cancer ?
    • What about the circulating tumour DNA (ctDNA) LUNAR test by Guardant Health designed to detect ctDNA in blood.

Should be evaluated upon submission of an in-depth revision of this article. 

Author Response

Dear reviewer! We appreciate greatly your kind attitude, careful analysis of the manuscript and a number of important recommendations for improvement. In accordance with your remarks we restructured the text, split the text into subchapters and made subheadings. We made a graphical scheme illustrating the process of searching, selecting and verifying markers suitable for diagnosis, prognosis, choice of therapy, evaluation of therapy effectiveness. The citations and references were checked carefully. We added information on the development and application of AI algorithms for minimally invasive CRC diagnostics. (Pages 2 and 25, the added text is underlined and colored yellow). We added information about the promising LUNAR test by Guardant Health designed to detect ctDNA in blood. (Page 13 the added text is underlined and colored yellow).

Reviewer 5 Report

The utilization of circulating markers in a liquid biopsy approach has proven to be highly effective in detecting molecular pathway alterations related to different types of cancer. In this review, the authors presented the key advancements in the field of circulating DNA and RNA-based markers for colorectal cancer (CRC). These markers include cancer-associated DNA mutations, DNA methylation changes, and shifts in noncoding RNA expression. The review focuses on the current circulating nucleic acid-based CRC markers, their potential application in clinical settings, and possibilities for future enhancements. The strategies employed for the discovery and validation of new markers were also discussed. The authors also addressed the existing challenges and proposing potential solutions. The specific comments to this manuscript are listed below.

Major
1. The correctness of references (refs) is important, especially for a review article. There are numerous errors in refs. and citations, which should be corrected. For examples, refs #6 and #7 did not have the information mentioned in the text (lines 47-51 and lines 66-69, respectively). Citations of refs# 155, 156 and 157 in Table 3 and in text (lines 764-769) were referred to different findings, likely due to incorrect numbering. The authors should have gone through the citations and references to ensure correctness.

2. A better reference for the content of lines 47-51 should be Virchows Arch (2016) 469:125–134.

3. Ref #28 is about non-small cell lung cancer diagnostic test, it may not suitable for the context of this manuscript. Also, the citation of ref #28 in lines 214-220 was not well justified.

4. Ref #62 (line 371-372 and Table 2) did not have the information referred to in the manuscript.

Minor

5. All abbreviations should be spelled out the first time appeared in the manuscript. For example, ddPCR in line 117 and CMS in line 50.

None

Author Response

Dear reviewer! We appreciate greatly your kind attitude and careful analysis of the manuscript. In accordance with your remarks, we restructured the text and corrected the citations and references. We changed the reference for the recommended Virchows Arch (2016) 469:125–134. We changed the text and citation and excluded Ref #28 which is about non-small cell lung cancer diagnostic test, (Page 4, the text is underlined and colored in yellow). We corrected the citation Ref #62 (line 371-372 and Table 2). The abbreviations have been added in text of article (the added abbreviations are colored green).

Round 2

Reviewer 3 Report

The manuscript still has relatively high similarity ratio (20%) compared to published literature. So the authors are urged to carry out substantial major revision to correct this unintentional plagiarism before any peer-review can be performed. The plagiarism check result for the revised manuscript is attached in this report.

Author Response

Dear Reviewer! We appreciate the thorough plagiarism check and did the best to make necessary corrections. We hope the revised manuscript improved. Would you be so kind as to analyze the revised manuscript plagiarism check result taking into account that some combination of words can’t be changed. For example, see line 67 «and fecal occult blood test (FOBT)»; line 83 “in the blood plasma and serum of cancer patients”; line 51 “comprising CMS1 (MSI immune), CMS2 (canonical), CMS3 (metabolic), and CMS4 (mesenchymal)”; line 160 “Commercial UltraSEEK platform (Agena Bioscience, San Diego, CA, United States)” and others.

The check made by the program gives many results of the words combinations, which are often used elsewhere.

Thank you in advance! Looking forward for your opinion about the manuscript.

Reviewer 4 Report

We have to acknowledge the authors of this review article for their extensive effort in data collection and analysis. However, this work is partly undermined by several flaws that have not yet been corrected. It is unfortunate because these flaws hinder the originality and dissemination of the messages. In short there is room for improvement.

My remarks are as follows:

  • Inappropriate references should be carefully identified and corrected. See for example ref 127 . This article is about non-small cell lung cancer. 
  • Ref 120 it is not about diagnosis of CRC. The data is very preliminary  (20 CRC patients) and its clinical significance is far from established. 
  • Ref 158 is puzzling. How can machine learning training from such a small sample lead to reliable predictions?
  • Several articles are cited as having diagnostic interest, even though the data is based on a very limited number of patients and controls, which obviously restricts the scope of interpretation. See for example ref 13 (cfDNA from 65 patients)  ref 131( 44 patients and 40 healthy controls), ref 141 (6 patients ), ref 156 (50 patients and 44 healthy controls).
  • The burden and standard of proof are very different across articles. The absence of predictive validation cohort creates problematic situations. Hence the informational content of article ref 193 that does no include a validation cohort (Based on 100 colorectal cancer (CRC) patients, 25 postoperative CRC patients, 28 colitis patients, and 108 healthy donors the authors conclude that Plasma hsa_circ_0006282 can be used as a novel diagnostic and dynamic monitoring biomarker in CRC patients) is very different from that of article ref 184 (see fig 8 that includes a predictive validation cohort (n=150)
  • Given the highly uneven value of the reports mentioned in this article, it is necessary to add in Tables 1, 3, and 4, a column which provides the minimum set of data required to assess their clinical relevance, i.e., the number of pathological cases vs. controls, as well as the presence or absence of a predictive validation cohort.

No comment

Author Response

Dear Reviewer! We appreciate the thorough analysis of the manuscript and did the best to make necessary corrections.

We checked the references, identified and corrected inappropriate ones.

  • Ref 127 in the list of refs which was about non-small cell lung cancer is changed, colored grey in the list of references.
  • Ref 158 seemed puzzling therefore we added the explanation given by the authors on the use of machine learning for the small sample in this case. (Chapter 1.2. Search for miRNA markers for CRC diagnostics) «Since the small sample size did not allow separate training and test samples, this problem was compensated by cross-validation».
  • We agree with your opinion that the burden and standard of proof are very different across articles. Therefore, we added in Tables 1, 3, and 4, a column which provides the data required to assess their clinical relevance, i.e., the number of pathological cases vs. controls, as well as the presence or absence of a predictive validation cohort.
  • We also added the following words into the Conclusion: “As we can see from the analysis of the cited reports, the diagnostic significance of the discovered markers is based on a limited number of patients and controls, as well as predictive validation cohort is rarely included.” Colored grey in the text.
  • Looking forward for your opinion about the revised manuscript.

Reviewer 5 Report

The revised manuscript only corrected the reference mistakes specified in the previous review without performing a thorough scrutiny of the references and citations. Numerous mistakes still present in the revised manuscript. Samples of errors remained in the revised manuscript are listed below.

1. Reference (ref) #24 does not contain information mentioned in the text (lines 222-224).

2. The numbering of citations were not in correct order. Citations of refs.# 32-34 appeared before refs.# 22-31 (line 206).

3. The sentence The study by Holm et al. [22] compared ddPCR, fully automated qRT-PCR–based system Idylla (Biocartis, Mechelen, Belgium), and NGS and demonstrated that Idylla displayed the sensitivity of at least the same level as ddPCR in the detection of the pre-selected KRAS mutations in the plasma cfDNA of mCRC subjects.” (lines 206-210) contained multiple mistakes. Ref.# 22, which did not contain the described study, also was not the work by Holm et al.

4. Citation for ref.# 23 did not exist in the manuscript.

5. Citation of ref.# 26 in Table 1 was incorrect, ref.#26 did not contain the described study.

6. Information of Ref. #27 was incorrect (Table 1), this study amplified 48 amplicons covering 240 key hotspot mutations of 14 genes (AKT1, APC BRAF, CTNNB1, EGFR, ERBB2, FBXW7, GNAS, KRAS, MAP2K1, NRAS, PIK3CA, MAD4, TP53). However, different from what was implied in the Table 1, not all these genes contained mutations.

7. The two citations (refs #26 and #27) between lines 332-336 were incorrect (likely switched). The descriptions in this paragraphs were inconsistent with the cited studies.

None

Author Response

Dear Reviewer! We appreciate the thorough analysis of the manuscript and did the best to make necessary corrections.

  • We checked the references, identified and corrected inappropriate ones.
  • Reference (ref) #24 didn’t not contain information mentioned in the text and it was removed from the text (Сhapter 2.3. Concordance between different platforms).
  • The numbering of citations was not in correct order. Citations of refs.# 32-34 appeared before refs.# 22-31 (line 206). This wrong order appeared due to the restructuring of the text in the process of the first revision, which led to the change of the Table 1 position in the text. The Table is moved into the proper position.
  • # 22 did not contain the described study, also was not the work by Holm et al. The ref was changed in the list of references.
  • Citation for ref.# 23 did not exist in the manuscript. We added description of the study in the text (Page 7, Chapter - 2.3. Concordance between different platforms).
  • Citation of ref.# 26 in Table 1 was incorrect, ref.#26 did not contain the described study. We added the text concerning this reference into the text (colored grey)
  • Information of Ref. #27 was incorrect (Table 1), this study amplified 48 amplicons covering 240 key hotspot mutations of 14 genes (AKT1, APC BRAF, CTNNB1, EGFR, ERBB2, FBXW7, GNAS, KRAS, MAP2K1, NRAS, PIK3CA, MAD4, TP53). However, different from what was implied in the Table 1, not all these genes contained mutations. We changed the number of genes in the Table 1
  • The two citations (refs #26 and #27) between lines 332-336 were incorrect (likely switched). The descriptions in this paragraph were inconsistent with the cited studies.
  • We corrected these citations, as well as others throughout the text and the Tables.
  • We hope the revised manuscript improved. Looking forward for your opinion about the revised manuscript.

Round 3

Reviewer 3 Report

In this review paper, the authors investigated the research trends in the discovery and application of circulating DNA and RNA based markers in diagnostic, therapeutic, and preventive strategies of colorectal cancer (CRC). The topic is very interesting. As the authors have corrected the unintentional plagiarism, the manuscript can be accepted for publication if the authors can carry out minor revision to discuss the pathogenesis of CRC further.

It is well established that programmed cell death like apoptosis is essential for health. Dysregulation of apoptosis promotes the development of different pathologies including CRC, and various inhibitor of apoptosis proteins (IAP) play critical role in the carcinogenesis of CRC [1]. Therapies counteract IAP activity and re-sensitize cancer cells to apoptosis may reduce both the severity of the disease and the CRC-related mortalities.

Attached please find the author keyword mapping of 2045 review papers relating inhibiting apoptosis with pathogenesis of CRC. There are 4733 Author Keywords in total, and the mapping captured 296 author keywords with occurrences of 5 times and above.

Reference:

1.       Cetraro, P.; Plaza-Diaz, J.; MacKenzie, A.; Abadía-Molina, F. A Review of the Current Impact of Inhibitors of Apoptosis Proteins and Their Repression in Cancer. Cancers 2022, 14, 1671. DOI: 10.3390/cancers14071671

Author Response

Dear Reviewer! We appreciate your valuable recommendation to add an important point into discussion of pathogenesis and the targeted therapeutic approaches in CRC. We agree that IAPs represent valuable therapeutic targets and potential predictive markers for the precise therapy selection. Please, find the text at the end of Chapter “2.3.1. CtDNA for the therapeutic decision” (highlighted grey).

Reviewer 4 Report

On reading this version I see no objection to the publication of this article. I recommend some minor modifications aimed at addressing a few inaccuracies that should be corrected by the authors through careful proofreading. For example "Line 275: "selection of adequate therapy with epidermal growth factor receptor (EGFR)": this wording is inaccurate (EGFR is not a therapy) and should be changed. In addition, the authors should cite the ESMO Scale for Clinical Actionability of Molecular Targets (ESCAT) in relation to ref [45]

Evaluation of this version is made very difficult 

Author Response

Dear Reviewer! We appreciate greatly your recommendation to add an important point into discussion of the predictive markers which have been supported as valuable for the selection of a targeted therapy in CRC. We added information about the scores of genetic alterations according to the ESCAT classification. Please, find the text in the Chapter “2.3.1. CtDNA for the therapeutic decision” (highlighted green).